# Mapping circuit dynamics during function and dysfunction

Srinivas Gorur-Shandilya[1], Elizabeth M Cronin[2], Anna C Schneider[2], Sara Ann Haddad[1†], Philipp Rosenbaum[1‡], Dirk Bucher[2], Farzan Nadim[2], Eve Marder[1*]

[1]Volen Center and Biology Department, Brandeis University, Waltham, United States; [2]Federated Department of Biological Sciences, New Jersey Institute of Technology and Rutgers University, Newark, United States

**Abstract** Neural circuits can generate many spike patterns, but only some are functional. The study of how circuits generate and maintain functional dynamics is hindered by a poverty of description of circuit dynamics across functional and dysfunctional states. For example, although the regular oscillation of a central pattern generator is well characterized by its frequency and the phase relationships between its neurons, these metrics are ineffective descriptors of the irregular and aperiodic dynamics that circuits can generate under perturbation or in disease states. By recording the circuit dynamics of the well-studied pyloric circuit in *Cancer borealis*, we used statistical features of spike times from neurons in the circuit to visualize the spike patterns generated by this circuit under a variety of conditions. This approach captures both the variability of functional rhythms and the diversity of atypical dynamics in a single map. Clusters in the map identify qualitatively different spike patterns hinting at different dynamic states in the circuit. State probability and the statistics of the transitions between states varied with environmental perturbations, removal of descending neuromodulatory inputs, and the addition of exogenous neuromodulators. This analysis reveals strong mechanistically interpretable links between complex changes in the collective behavior of a neural circuit and specific experimental manipulations, and can constrain hypotheses of how circuits generate functional dynamics despite variability in circuit architecture and environmental perturbations.

*For correspondence:
marder@brandeis.edu

Present address: †Department of Molecular Life Sciences, University of Zürich, Zürich, Switzerland; ‡GlobalData, New York, United States

Competing interest: The authors declare that no competing interests exist.

## Editor's evaluation

This study applies an unsupervised dimensionality reduction (t-SNE) to characterize neural spiking dynamics in the pyloric circuit in the stomatogastric ganglion of the crab, an important system for mechanistic analysis of rhythmic circuit function. The application of unsupervised methods to characterize qualitatively distinct regimes of spiking neural circuits is interesting and novel. The challenges and lessons learned in this study are of broader interest to those seeking to quantitatively characterize large sets of neural data across many subjects. The method is demonstrated across hundreds of animal subjects and used to investigate circuit responses to a variety of perturbations.

## Introduction

Neural circuits can generate a wide variety of spiking dynamics, but must constrain their dynamics to function appropriately. Cortical circuits maintain irregular spiking patterns through a balance of excitatory and inhibitory inputs (*van Vreeswijk and Sompolinsky, 1996*; *Mariño et al., 2005*; *Brunel and Wang, 2003*) and the loss of canonical dynamics is associated with neural diseases like channelopathies and epilepsy (*Marbán, 2002*; *Staley, 2015*). Preserving functional dynamics can be a challenge

for neural circuits for the following reasons. The same spike pattern can be generated by diverse circuits with many different topologies and broadly distributed synaptic and cellular parameters (*Prinz et al., 2004*; *Golowasch et al., 2002*; *Alonso and Marder, 2019*; *Memmesheimer and Timme, 2006*). Furthermore, neural circuits are constantly being reconfigured, with ion channel protein turnover, and homeostatic feedback mechanisms modifying conductance and synapse strengths continuously (*Turrigiano et al., 1994*; *Turrigiano et al., 1995*; *O'Leary et al., 2014*; *Franci et al., 2020*). The problem of maintaining functional activity patterns is aggravated by the fact that functional circuit dynamics tend to lie within a low-dimensional subspace within the high-dimensional state space: of the numerous possible solutions, only a few are functional and are found in animals (*Cunningham and Yu, 2014*; *Pang et al., 2016*). How do neural circuits preserve functional dynamics despite these obstacles?

Answering this question requires, as a prerequisite, a quantitative description of the dynamics of neural circuits during function and dysfunction. When rhythms are regular, this is relatively simple, but when rhythms become irregular, classifying them becomes hard (*Haddad and Marder, 2018*; *Tang et al., 2012*; *Haley et al., 2018*). In this article, we study the dynamics of a well-studied central pattern generator, the pyloric circuit in the stomatogastric ganglion (STG) in *Cancer borealis* (*Marder and Bucher, 2007*). The pyloric circuit is small, in crabs consisting of 13 neurons coupled by inhibitory and electrical synapses. Its topology and cellular dynamics are well understood, and the circuit generates a clearly defined 'functional' collective behavior where bursts of spikes from three different cell types alternate rhythmically to generate a triphasic motor pattern. The stereotypy and periodicity of the motor pattern suggest that the baseline dynamics of the pyloric circuit are fundamentally low dimensional. This has allowed for the effective parameterization of the rhythm by a small number of ad hoc descriptors such as the burst period, duty cycles, and phase of each neuron (*Hartline and Maynard, 1975*; *Eisen and Marder, 1984*; *Miller and Selverston, 1982*).

In response to perturbations that span many cycles, pyloric circuit dynamics are not always periodic, and descriptors that work well to characterize the canonical rhythm are inadequate to describe these atypical dynamic states. Efforts to study circuit dynamics under these regimes, and to characterize how the circuit responds to, and recovers from perturbations, have been frustrated by the inability to quantitatively describe irregular and non-stationary dynamics (*Haddad and Marder, 2018*; *Tang et al., 2012*; *Haley et al., 2018*).

In this article, we set out to address the problem of quantitatively describing neural circuit dynamics under a variety of conditions. We reasoned that circuit dynamics lie on some lower-dimensional set within the full high-dimensional space of possible dynamics, even when circuits exhibit atypical and nonfunctional behavior, because even circuits generating dysfunctional dynamics are still constrained by cellular parameters and network topology. We therefore set out to find and visualize this subset of spike patterns using an unsupervised machine learning approach to visualize patterns in the high-dimensional data in two dimensions. This method allows us to visualize the totality of a large and complex dataset of spike patterns, while being explicit about the assumptions and biases in the analysis. Using this method, we found nontrivial features in the distribution of the data that hinted at diverse, stereotyped responses to perturbations. Using this compact representation allowed us to efficiently manually classify these patterns and measure transitions between these patterns. We were thus able to characterize the diversity of circuit dynamics under baseline and perturbed conditions, and identify anecdotally observed atypical states within the full repertoire of spiking patterns for many hundreds of animals.

## Results

### Perturbations can destabilize the triphasic pyloric rhythm

Studies that measure the pyloric rhythm commonly involve recording from nerves from the STG in ex vivo preparations. Preparations typically also include the stomatogastric nerve (*stn*) that carries the axons of descending neuromodulatory neurons from the esophageal and commissural ganglia that project into the STG. Under baseline conditions (11°C, with the *stn* intact, *Figure 1a*), the periodic triphasic oscillation of the pyloric circuit can be measured by extracellular recordings of the lateral pyloric, pyloric dilator, and pyloric nerves (*lpn*, *pdn*, and *pyn*) (*Figure 1a*). Bursts of spikes from the pyloric dilator (PD) neurons on the *pdn* are followed by bursts of spikes from the lateral pyloric neuron

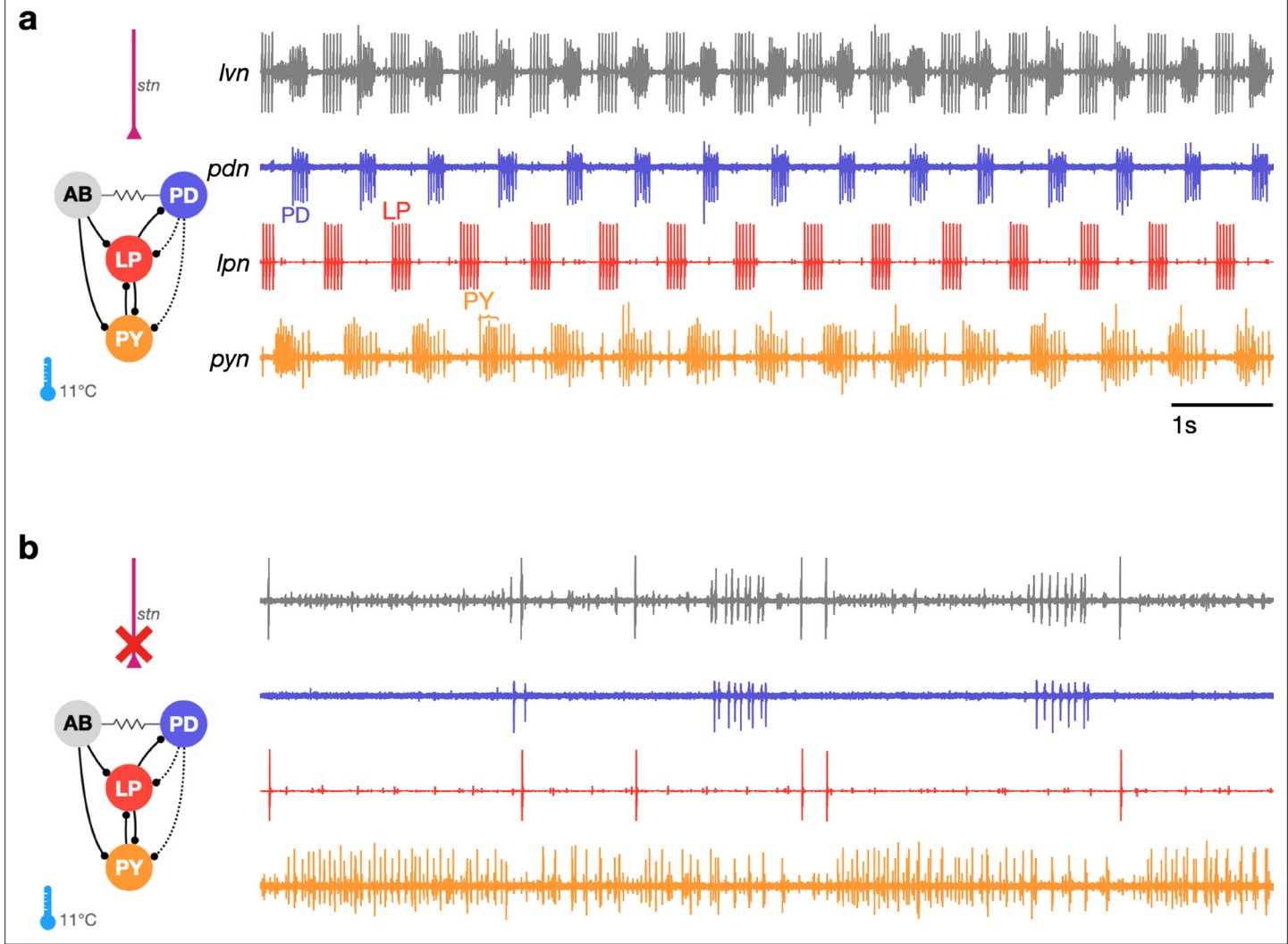

**Figure 1.** The triphasic pyloric rhythm can become irregular and hard to characterize under perturbation. (**a**) Simplified schematic of part of the pyloric circuit (left). Filled circles indicate inhibitory synapses, solid lines are glutamatergic synapses, and dotted lines are cholinergic synapses. Resistor symbol indicates electrical coupling. The pyloric circuit is subject to descending neuromodulatory control from the stomatogastric nerve (*stn*). Right: simultaneous extracellular recordings from the *lvn*, *lpn*, *pdn,* and *pyn* motor nerves. Action potentials from lateral pyloric (LP), pyloric dilator (PD), and pyloric (PY) are visible on *lpn*, *pdn*, and *pyn*. Under these baseline conditions, PD, LP, and PY neurons burst in a triphasic pattern. The anterior burster (AB) neuron is an endogenous burster and is electrically coupled to PD neurons. (**b**) When the *stn* is cut, neuromodulatory input is removed and the circuit is 'decentralized.' In this case, the pyloric rhythm can become irregular and hard to characterize. In addition, spikes from multiple PY neurons can become harder to reliably identify on *pyn*.

(LP) on *lpn* and bursts of spikes from the pyloric neurons (PY) on *pyn*. Spikes from lateral posterior gastric (LPG) neurons are also found on the *pyn* nerve in these recordings and can be differentiated from PY spikes by their shape and timing (LPG is active during PD bursts). Under these control conditions, where the rhythm is robust and spikes from these neurons are easily identifiable both by their location on the nerve and their phase in the cycle, the dual problems of identifying spikes from raw extracellular recordings and meaningfully describing circuit dynamics are easily resolvable.

In studies that characterize the changes in circuit dynamics to prolonged perturbations, spike identification and circuit dynamics characterization are less straightforward. For example, when descending neuromodulatory projections from the *stn* are cut (i.e., when the STG is decentralized, *Figure 1b*), the collective dynamics of the pyloric circuit can become less regular. This loss of regularity is concomitant with spikes being harder to reliably identify in extracellular recordings. While PD and LP neuron spikes can still be typically easily identified on the *pdn* and *lpn* nerves (*Figure 1b*), identifying PY on the *pyn* in the absence of a regular rhythm can be challenging. This problem is aggravated by the fact that

spikes from the LPG neuron are frequently found on *pyn*, and because there are several copies of the PY neuron, whose spikes can range from perfect coincidence to slight offsets that can unpredictably change the amplitude and shape of PY spikes due to partial summation. For these reasons, some previous works studying the response of pyloric circuits to perturbations have consistently recorded from the *lpn* and *pdn* nerves, but not from the *pyn* (*Hamood et al., 2015*; *Haley et al., 2018*; *Haddad and Marder, 2018*; *Rosenbaum and Marder, 2018*). Therefore, in order to include the largest number of experiments in our analysis, we chose to characterize the dynamics of the LP and PD neurons.

## Nonlinear dimensionality reduction allows for the visualization of diverse pyloric circuit dynamics

The regular pyloric rhythm involves out-of-phase bursts of spikes between LP and PD, and is observed under baseline conditions (*Figure 2a1-3*). Perturbations such as the removal of descending neuro-modulatory inputs, changes in temperature, or changes in pH can qualitatively alter the rhythm, leading to a large variety of hard-to-characterize spiking patterns (*Figure 2a4-6*). Because these irregular states may lose the strong periodicity found in the canonical motor pattern, burst metrics such as burst period or phase offsets between bursts that work well to characterize the regular rhythm perform poorly. Efforts to characterize and quantify these atypical spike patterns must overcome the slow timescales in observed dynamics, the large quantity of data, and irregularity and variability in observed spike trains. Previous work used ad hoc categorization systems to assign observations of spike trains into one of a few groups (*Haddad and Marder, 2018*; *Haley et al., 2018*), but these categorization methods scaled poorly and relied on subjective annotations.

We sought instead to visualize the totality of pyloric circuit dynamics under all conditions using a method that did not rely on a priori identification of (non)canonical dynamic patterns. Such a data visualization method, while descriptive, would generate a quantitative vocabulary to catalog the diversity of spike patterns observed both when these patterns were regular and also when they were irregular and aperiodic, thus allowing for the quantitative characterization of data previously inaccessible to traditional methods (*Börner et al., 2005*; *Nguyen and Holmes, 2019*).

The visualization was generated as follows: time-binned spike trains were converted into their equivalent interspike interval (ISI) and phase representations (*Figure 2b*, Materials and methods). For all analyses, we consider nonoverlapping 20 s time bins. We chose this time bin following inspection of circuit dynamics across many conditions in several animals. Because there can be an arbitrary number of spikes in a bin, there are an arbitrary number of ISIs and phases. This makes it challenging to find a basis to represent the entire dataset. Ideally, we want to represent the spike pattern in each 20 s bin with a point in some space of high but fixed dimensionality. To convert this into a vector of fixed length, we measured percentiles of ISIs and phases (*Figure 2c*). Together with other metrics (like ratios of ISIs, measures that capture discontinuities in ISI distributions, see Materials and methods for details), these percentiles were assembled into a fixed-length vector and each dimension was $z$-scored across the entire dataset (*Figure 2d*). A collection of spike trains from an arbitrary number of neurons has thus been reduced to a matrix where each row consists of $z$-scored percentiles of ISIs and other metrics. This matrix can be visualized using a nonlinear dimensionality reduction technique such as $t$-distributed stochastic neighbor embedding (t-SNE) (*Van der Maaten and Hinton, 2008*), which can generate a two-dimensional representation of the full dataset (*Figure 2e*).

In this representation, each dot corresponds to a single time bin of spike trains from both neurons. We found by manual inspection that spike trains that are visually similar (*Figure 2a1-3*) tend to occur close to each other in the embedding (*Figure 2e1-3*). Spike patterns that are qualitatively different from each other (*Figure 2a4-6*) tended to occur far from each other, often in clusters separated by regions of low data density (*Figure 2e4-6*, *Supplementary file 1*).

How useful is such a visualization and does it represent the variation in spike patterns in the data in a reasonable manner? We colored each point by classically defined features such as the burst period or the phase (*Figure 2—figure supplement 1*). We found that the embedding arranges data so that differences between clusters and within clusters had interpretable differences in various burst metrics. For example, clusters on the left edge of the map tended not to have defined LP phases, typically due to silent or very sparse LP firing (*Figure 2—figure supplement 1b*). Location of data in the largest cluster was correlated to firing rate in the PD neuron (*Figure 2—figure supplement 1c*). We observed that burst metrics, when they were defined, tended to vary smoothly across the map. To quantify this

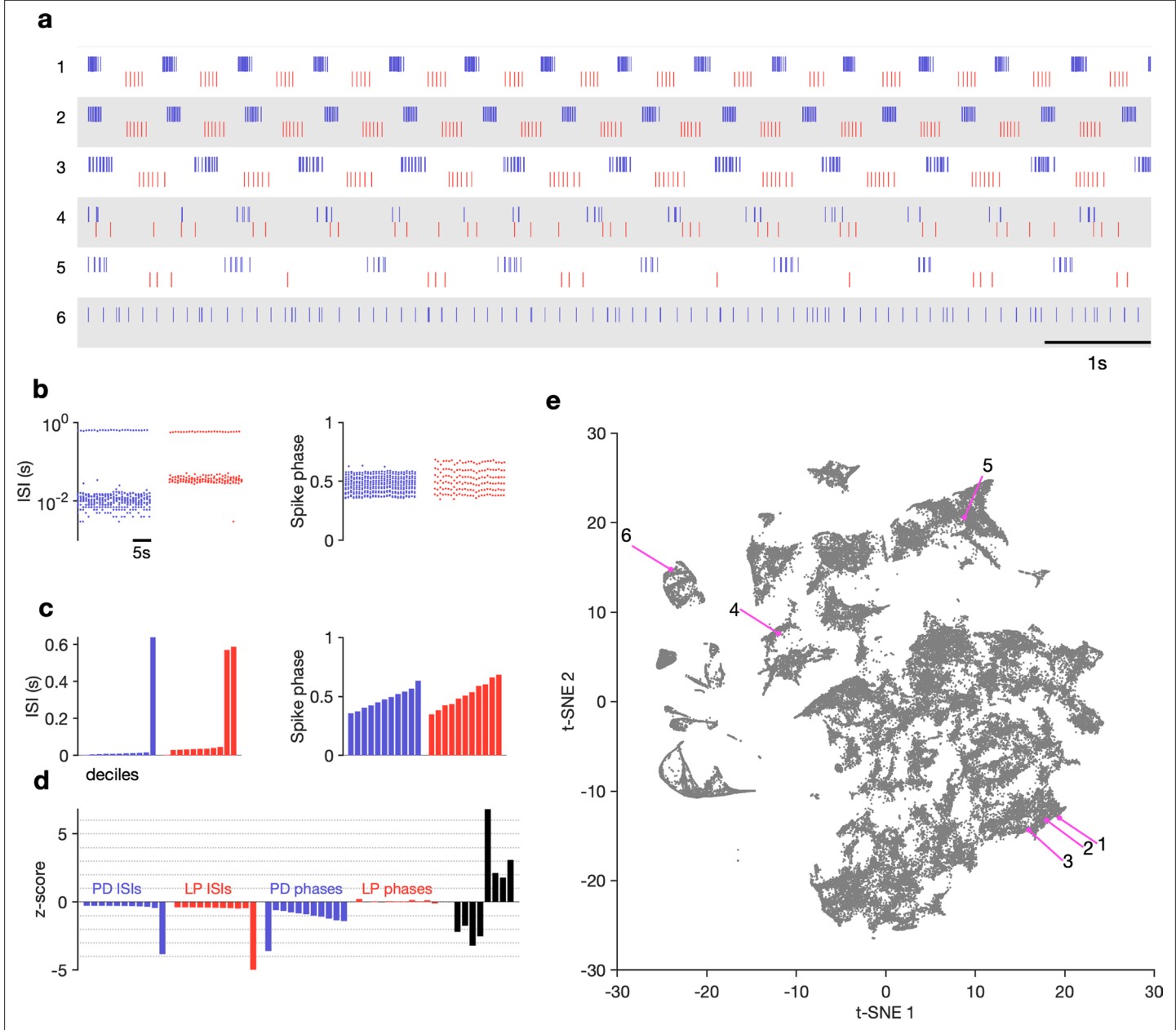

**Figure 2.** Visualization of diverse neural circuit dynamics. (**a**) Examples of canonical (1–3) and atypical (4–6) spike patterns of pyloric dilator (PD; blue) and lateral pyloric (LP; red) neurons. Rasters show 10 s of data. (**b–d**) Schematic of data analysis pipeline. (**b**) Spike rasters in (**a2**) can be equivalently represented by interspike intervals (ISIs) and phases. 20 s bins shown. Each 20 s bin contains a variable number of spikes/ISIs. (**c**) Summary statistics of ISI and phase sets in (**d**), showing tenth percentiles. Using percentiles converts the variable length sets in (**b**) to vectors of fixed length. (**d**) $z$-scored data assembled into a single vector, together with some additional measures (Materials and methods). (**e**) Embedding of data matrix containing all vectors such as the one shown in (**d**) using $t$-distributed stochastic neighbor embedding (t-SNE). Each dot in this image corresponds to a single 20 s spike train from both LP and PD. Example spike patterns shown in (**a**) are highlighted in the map. $n = 94,844$ points from $N = 426$ animals. In (**a–d**), features derived from LP spike times are shown in red, and features derived from PD spike times are shown in blue.

The online version of this article includes the following figure supplement(s) for figure 2:

**Figure supplement 1.** Burst metrics smoothly vary in map.

**Figure supplement 2.** Principal components analysis (PCA) and k-means to find clusters in feature vectors.

**Figure supplement 3.** Embedding arranges data so that neighbors tend to be similar.

**Figure supplement 4.** Effect of varying perplexity in $t$-distributed stochastic neighbor embedding (t-SNE) embedding.

**Figure supplement 5.** Validation of method using synthetic data.

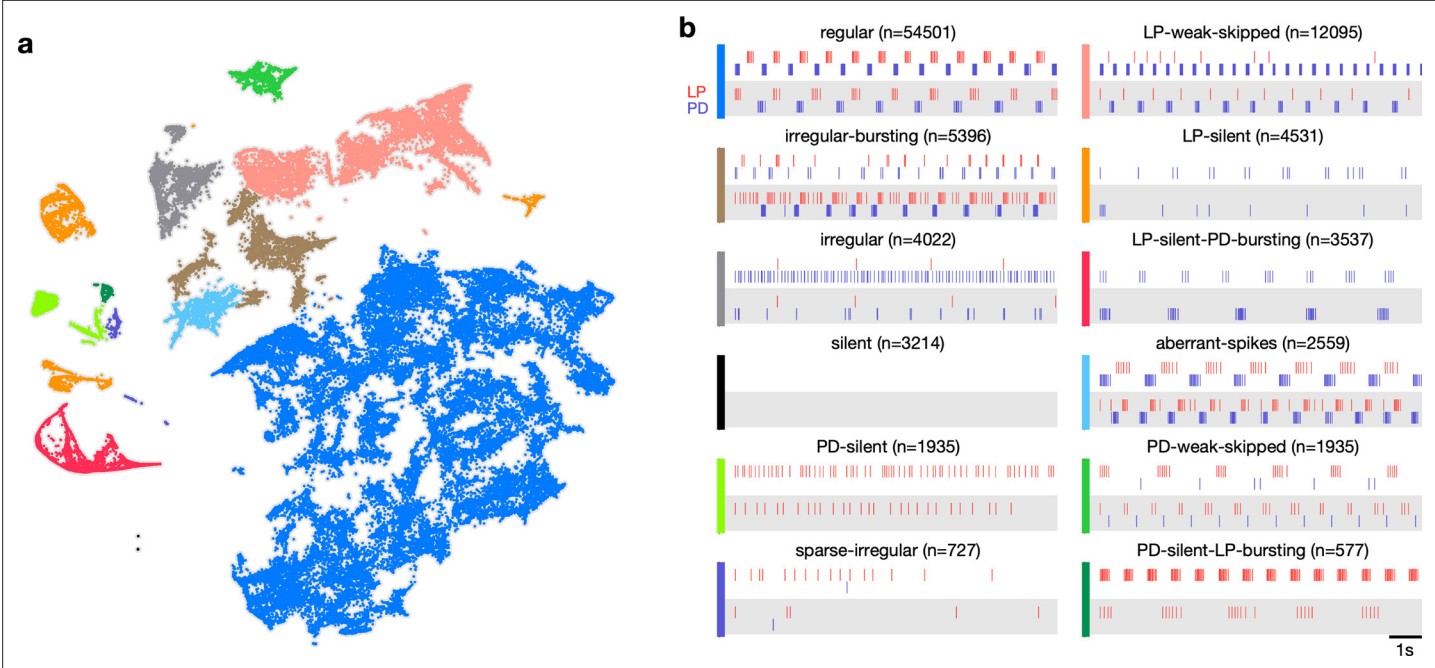

**Figure 3.** Map allows identification of distinct spiking dynamics. (**a**) Map of all pyloric dynamics in dataset where each point is colored by manually assigned labels. Each point corresponds to a 20 s paired spike train from lateral pyloric (LP) and pyloric dilator (PD) neurons. Each panel in (**b**) shows two randomly chosen points from that class. The number of points in each class is shown in parentheses above each panel. $n = 94,844$ points from $N = 426$ animals. Labels are ordered by likelihood in the data.

The online version of this article includes the following figure supplement(s) for figure 3:

**Figure supplement 1.** Speed of trajectories through map.

**Figure supplement 2.** Embeddings with different initializations.

**Figure supplement 3.** Using Uniform Manifold Approximation and Projection (uMAP) instead of *t*-distributed stochastic neighbor embedding (t-SNE).

observation, we built a Delaunay triangulation (Materials and methods) on the embedded data and measured the triadic differences between PD burst periods and PD duty cycles (*Figure 2—figure supplement 3*). Triadic differences in these metrics were significantly smaller in the map than triadic differences in a projection of the first two principal components or a shuffled map (p<0.0001, Kolmogorov–Smirnv test), suggesting that the t-SNE cost function generates a useful embedding where spike features vary smoothly within clusters. Finally, to validate our approach, we generated a synthetic dataset with different classes of spike patterns (Materials and methods) and analyzed it similarly. Coloring points in the t-SNE embedding by the original class revealed that clusters in the t-SNE map corresponded to different classes in the synthetic data, suggesting that this method can identify and recover stereotyped spike patterns in neural data (*Figure 2—figure supplement 5*).

## Visualization of circuit dynamics allows manual labeling and clustering of data

Previous studies have shown that regular oscillatory bursting activity of the pyloric circuit can qualitatively change on perturbation. Circuit dynamics can be highly variable and has been categorized into various states such as 'atypical firing,' 'LP-01 spikes,' or 'atypical' (*Haddad and Marder, 2018*; *Haley et al., 2018*). Both the process of constructing these categories and the process of classifying data into these categories are typically done manually, and therefore requires expert knowledge that is not explicitly captured and is impossible to reproduce. Because the embedding distributed data into clusters, we hypothesized that clusters corresponded to stereotyped dynamics that were largely similar, and different clusters represented the qualitatively different circuit dynamics identified by earlier studies.

We therefore manually inspected circuit dynamics at randomly chosen points in each apparent cluster and generated labels to describe the dynamics in that region (*Figure 3*). This process colored

the map and segmented it into distinct regions that broadly followed, and were largely determined by, the distribution of the data in the embedding (*Figure 3a*). Most of the data (57%) were assigned the regular label, where both PD and LP neurons burst regularly in alternation with at least two spikes per burst, and all identified regular states occurred in a single contiguous region in the map (blue). In the LP-weak-skipped state, PD bursts regularly, but LP does not burst every cycle, or only fires a single spike per burst. Irregular-bursting states showed bursting activity on both neurons, which were interrupted or otherwise irregular. In contrast, the irregular state showed spiking that was more variable and did not show strong signs of bursting at any point. LP-silent-PD-bursting states had regular bursting on PD, with no spikes on LP, while LP-silent states also had no spikes on LP, but activity on PD was more variable, and did not show regular bursting.

The time evolution of the pyloric dynamics of every preparation constitutes a trajectory in the map, and every point in the map is therefore associated with an instantaneous speed of motion in the map. We hypothesized that instantaneous speed could vary across the map, with points labeled regular moving more slowly through the map than points with labels corresponding to atypical states such as irregular because regular rhythms would vary less over time. Consistent with this, we found that points in the regular cluster tended to have smaller speeds than points in other clusters (*Figure 3—figure supplement 1a*). Speeds in the regular state were significantly lower than every other state except PD-silent-LP-bursting (p<0.004, permutation test), suggesting that atypical states were associated with increased variability in circuit dynamics (*Figure 3—figure supplement 1b*).

Do the clusters we see in the data, and the resultant categorization of the data, depend strongly on the details of the dimensionality reduction method we used (t-SNE)? We used an entirely different embedding algorithm (Uniform Manifold Approximation and Projection [uMAP], *McInnes et al., 2018*) to embed the feature vectors in two-dimensional space. The map generated by uMAP preserved the coarse feature of the t-SNE embedding, suggesting that the features in the map reflected the features of the distribution of the data more strongly than details of the dimensionality reduction method. Coloring points in the uMAP embedding (*Figure 3—figure supplement 3*) revealed a roughly similar organization of data in the embedding space, suggesting that our categorization method did not strongly depend on the details of the dimensionality reduction.

## Variability in baseline circuit dynamics across a population of wild-caught animals

Work on the pyloric circuit has used a wild-caught crustacean population. This uncontrolled environmental and genetic variability serves as a window into the extant variability of a functional neural circuit in a wild population of animals. In addition, experimental and computational work has shown that similar rhythms can be generated by a wide variety of circuit architectures and cellular parameters (*Prinz et al., 2003*; *Hamood and Marder, 2014*; *Alonso and Marder, 2019*). We therefore set out to study the variability in baseline circuit dynamics in the 346 pyloric circuits recorded under baseline conditions in this dataset.

The burst period of the pyloric circuit in the lobster can vary two- to threefold under baseline conditions at 11°C across animals (*Bucher et al., 2005*). Despite this sizable variation, other burst metrics, such as the phase onset of follower neurons, or the duty cycles of individual neurons, are tightly constrained (*Bucher et al., 2005*), likely related to the fact that these circuits are under activity-dependent feedback regulation (*Turrigiano et al., 1995*; *O'Leary et al., 2014*; *Gorur-Shandilya et al., 2020*) as they develop and grow. Activity-dependent regulation of diverse pyloric circuits could constrain variability in a single circuit across time to be smaller than variability across the population.

To test this hypothesis, we measured a number of burst metrics such as burst period and the phases and duty cycles of the two neurons across these 346 preparations in baseline conditions (*Figure 4*) when data are labeled regular because metrics are well-defined in this state. The mean values of each of these metrics were unimodally distributed (*Figure 4a*) and the coefficient of variation (CV) for all metrics was approximately 0.1 (*Figure 4b*). Using the mean CV in each individual as a proxy for the within-animal variability, and the CV of the individual means as a proxy for the across-animal variability, we found that every metric measured was more variable across animals than within animals (*Figure 4c*). Shuffling experimental labels generated null distributions for excess variability across animals and showed that across-animal variability was significantly greater than within-animal variability (*Figure 4d*, p<0.007, permutation test, *Table 1*).

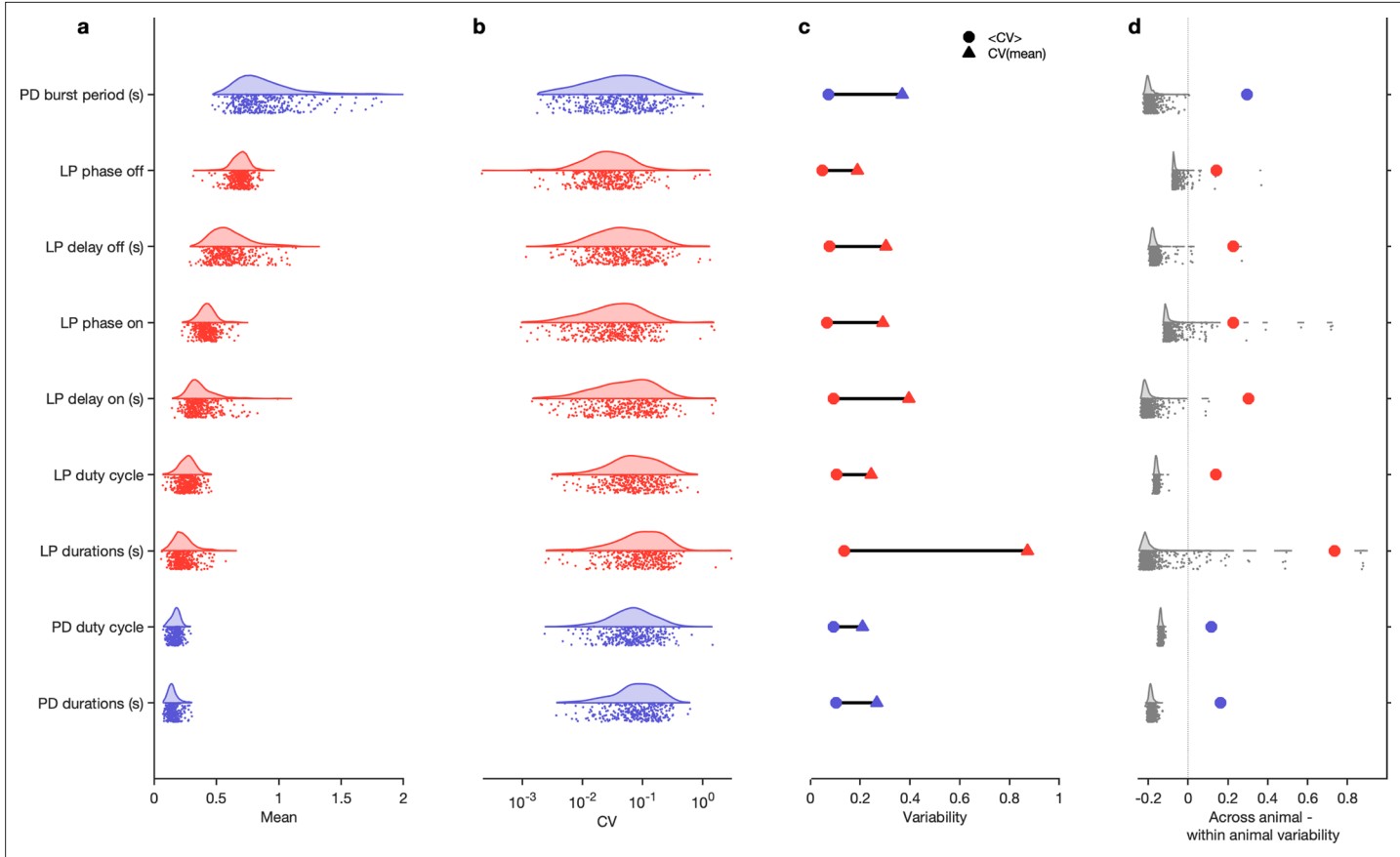

**Figure 4.** Variability of burst metrics under baseline conditions. (**a**) Variability of burst metrics in pyloric dilator (PD) and lateral pyloric (LP) neurons across a population of wild-caught animals. Metrics are only computed under baseline conditions and in the regular cluster. (**b**) Distribution of coefficient of variation (CV) of metrics in each animal across all data from that animal. In (**a, b**), each dot is from a single animal, and distributions show variability across the entire population. (**c**) Across-animal variability (CV of individual means, Δ) is greater than within-animal variation (mean of CV in each animal, O) for every metric. (**d**) Difference between across-animal variability and within-animal variability (colored dots). For each metric, gray dots and distribution show differences between across-animal and within-animal variability for shuffled data. $n = 18,336$ points from $N = 346$ animals.

The online version of this article includes the following figure supplement(s) for figure 4:

**Figure supplement 1.** State distribution under baseline conditions.

**Figure supplement 2.** Recording condition alters regular state probability.

**Figure supplement 3.** Effect of sea surface temperature on baseline circuit dynamics.

It is reasonable to suppose that all baseline data exist in the regular cluster. While most baseline data are confined to the regular cluster ($\approx 80\%$, *Figure 4—figure supplement 1a*), the remaining data, nominally recorded under baseline conditions, contains atypical circuit dynamics (*Figure 4—figure supplement 1b and c*). What causes these atypical circuit dynamics in this large, unbiased survey of baseline pyloric activity? One possibility could be inadvertent damage to the preparation caused by dissection and preparation of the circuit for recording. Consistent with this, we found that the probability of observing regular states was significantly reduced when cells were recorded from intracellularly (*Figure 4—figure supplement 2*), which may be due to increase in leak currents owing to impaling cells with sharp electrodes (*Cymbalyuk et al., 2002*) or due to cell dialysis (*Hooper et al., 2015*). No significant correlation was observed between sea surface temperatures (a proxy for environmental conditions for these wild-caught animals) and burst metrics (*Figure 4—figure supplement 3a–c*) or the probability of observing a regular state (*Figure 4—figure supplement 3d*). Taken together, these results underscore the importance of verifying that baseline or control data does not include uncontrolled technical variability that could mask biological effects of interest.

**Table 1.** ANOVA results and power analysis for *Figure 4*.

ANOVA results for burst metrics in baseline conditions. For each metric, each animal is treated as a group and the variability (mean square difference) is compared within and across group. $F$ is the ratio of across-animal to within-animal mean square differences. $N_{.99}$ is the estimate of the sample size required to reject the null hypothesis with a probability of 0.99 when the alternative hypothesis is true. $N = 346$ animals.

| Metric | Across-animal MS | Within-animal MS | F | $N_{.99}$ |
|---|---|---|---|---|
| LP delay off (s) | 1.1391 | 0.010 956 | 103.97 | 6 |
| LP delay on (s) | 0.616 47 | 0.0111 | 55.54 | 6 |
| LP durations (s) | 0.363 86 | 0.012 366 | 29.424 | 4 |
| LP duty cycle | 0.159 86 | 0.001 309 3 | 122.09 | 10 |
| LP phase off | 0.234 06 | 0.007 227 9 | 32.383 | 11 |
| LP phase on | 0.216 55 | 0.008 811 5 | 24.576 | 9 |
| PD burst period (s) | 3.557 | 0.036 872 | 96.469 | 4 |
| PD durations (s) | 0.079 397 | 0.000 549 44 | 144.5 | 6 |
| PD duty cycle | 0.053 472 | 0.000 413 23 | 129.4 | 16 |

LP: lateral pyloric; PD: pyloric dilator. MS: mean square.

## Perturbation modality alters state probability

The pyloric circuit and other circuits in the crab must exhibit robustness to the environmental perturbations that these animals are likely to encounter. Previous studies have characterized the ability of crustacean circuits to be robust to environmental perturbations such as pH (*Haley et al., 2018*; *Ratliff et al., 2021*; *Qadri et al., 2007*), temperature (*Tang et al., 2010*; *Tang et al., 2012*; *Rinberg et al., 2013*; *Haddad and Marder, 2018*; *Kushinsky et al., 2019*), oxygen levels (*Clemens et al., 2001*), and changes in extracellular ionic concentrations (*He et al., 2020*). Robustness to these perturbations exists up to a limit, likely reflecting the bounds of the natural variation in these quantities that these circuits are evolved to function in. When challenged with extremes of any of these perturbation modalities, the pyloric rhythm breaks down, displaying irregular or aberrant states, and may even cease spiking entirely. Such states are commonly referred to as 'crashes' and can have many flavors (*Haddad and Marder, 2018*; *Tang et al., 2010*; *Tang et al., 2012*) and involve the loss of the characteristic antiphase activity in the LP and PD neurons.

It remains unclear if extreme perturbations of different modalities share common pathways of destabilizing and disrupting the pyloric rhythm (*Ratliff et al., 2021*). In principle, these environmental perturbations can disrupt neuron and circuit function in qualitatively different ways: for example, changes in extracellular potassium concentration can alter the reversal potential of potassium (*He et al., 2020*) vs. changes in temperature can have varied effects on the timescales and conductances of all ion channels (*Tang et al., 2010*; *Caplan et al., 2014*). Because prior work was focused on studying the limits of robustness and lacked a detailed quantitative description of irregular behavior, the fine structure of the transition between functional dynamics and silent or 'crashed' states remain poorly characterized (*Ratliff et al., 2021*). We therefore set out to measure how pH, temperature, and extracellular potassium perturbations alter circuit state probability.

Where in the map are data under extreme environmental perturbations? Circuit spike patterns under high pH (>9.5), high temperature (>25°C), and high extracellular potassium ($2.5 \times [K^+]$) are distributed across a wide region of the map, spanning both regions in the regular cluster and other nonregular clusters (*Figure 5a*). Spike patterns observed under high-temperature conditions in the regular region were clustered in the lower extremity, in the region containing high firing rates and small burst periods of PD (*Figure 2—figure supplement 1*), consistent with earlier studies showing that elevated temperatures tend to speed up the pyloric rhythm (*Tang et al., 2010*; *Tang et al., 2012*).

Subjecting the pyloric circuit to extremes of pH, temperature, and extracellular potassium altered the distribution of observed states (*Figure 5c*). In all cases, the probability of observing regular was

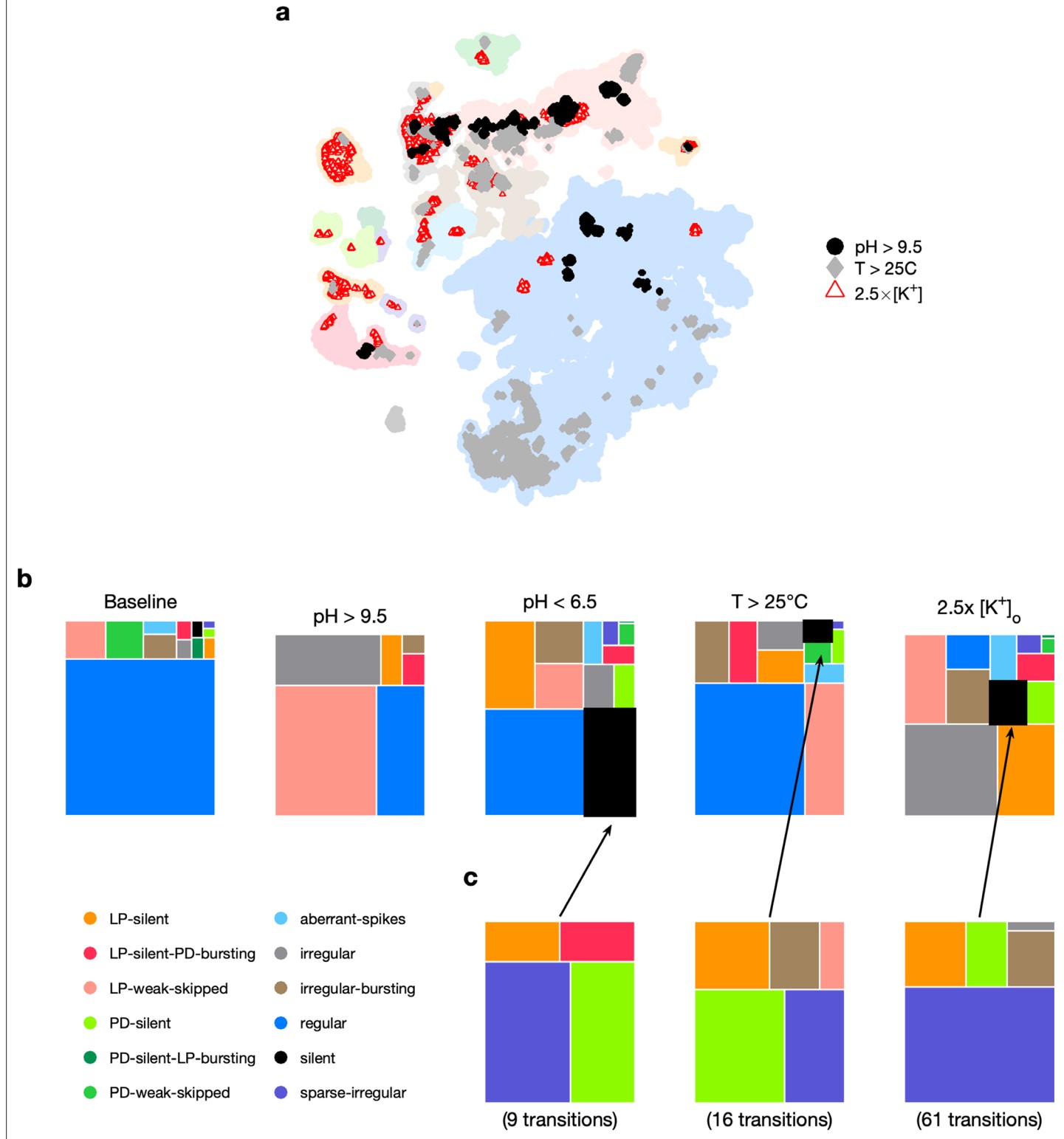

**Figure 5.** Effect of three different environmental perturbations. (**a**) Map showing regions that are more likely to contain data recorded under extreme environmental perturbations. (**b**) Treemaps showing probability distributions of states under baseline and perturbed conditions. (**c**) Probability distribution of states preceding silent state under perturbation. pH perturbations: $n = 4023$ from 6 animals; $[K^+]$ perturbations: $n = 5526$ from 20 animals; temperature perturbations: $n = 80,470$ from 414 animals.

The online version of this article includes the following figure supplement(s) for figure 5:

**Figure supplement 1.** Preparation-by-preparation response to pH perturbations.

significantly reduced (p<0.001, paired permutation test), and a variety of nonregular states were observed. We observed that high pH (>9.5) did not silence the preparation, but silent states were observed in low pH (<6.5), consistent with previously published manual annotation of this data (*Haley et al., 2018*). Silent states were also observed in $2.5 \times [K^+]$, as reported earlier by *He et al., 2020*. Previous work has shown that the isolated pacemaker kernel (AB and PD neurons) has a stereotyped trajectory from bursting through tonic spiking to silence when subjected to pH perturbations (*Ratliff et al., 2021*), but moves through a different trajectory (bursting to weak bursting to silence) during temperature perturbations. Do pathways to silent states share similarities across perturbation modality in intact circuits? To answer this, we plotted the probability of observing states conditioned on the transition to silence in low pH, high temperature, and $2.5 \times [K^+]$ (*Figure 5d*). In the $\approx 2000$ transitions between states detected, we never observed a transition from regular to silent, suggesting that the timescales of silencing are slow, longer than the width of one data bin (20 s). Trajectories to silent states always transition through a few intermediate states such as `sparse-irregular`, `LP-silent`, or `PD-silent` (*Figure 5d*).

## Transitions between states during environmental perturbations

Changes in temperature, pH, and $[K^+]$ have different effects on the cells in the pyloric circuit and therefore can destabilize the rhythm in different ways. Increasing the extracellular $[K^+]$ changes the reversal potential of $K^+$ ions, altering the currents flowing through potassium channels, and typically depolarizes the neuron (*He et al., 2020*). Ion channels can be differentially sensitive to changes in temperature or pH, and changes in these variables can have complex effects on ionic currents in neurons (*Tang et al., 2010*; *Tang et al., 2012*; *Haley et al., 2018*). We therefore asked if different environmental perturbations changed the way in which regular rhythms destabilized.

Our analysis mapped a time series of spike times from PD and LP neurons to a categorical time series of labels such as regular. We therefore could measure the transitions between states during different environmental perturbations (Materials and methods). We found that transition matrices between states shared commonalities across environmental perturbations (*Figure 6a*), such as likely transitions between regular and LP-weak-skipped states. PD-silent-LP-bursting states tended to be followed by PD-silent states, in which the LP neuron is spiking, but not bursting regularly. The LP neuron becomes less regular in both transitions, contributing to the loss of regular rhythms. We never observed a transition from regular rhythms LP-silent or PD-silent states, suggesting slow (>20 s) timescales of rhythm collapse. In high pH, every transition away from the regular state was to the LP-weak-skipped state, hinting at increased sensitivity of the LP neuron to high pH. High pH perturbations also never silenced the circuit, as previously reported (*Haley et al., 2018*), and showed fewer and less varied transitions than other perturbations. Are some transitions over- or underrepresented in the transition matrix? To determine this, we constructed a null model where transitions occurred with probabilities that scaled with the marginal probability of final states (Materials and methods). Transitions that occurred significantly more often than predicted by the null model are shown with black borders and those that occurred significantly less often than predicted are shown with filled circles (*Figure 6a*). Transitions that never occurred but occurred at significantly nonzero rates in the null model are indicated with diamonds.

Earlier work has shown that transitions from regular bursting are preceded by an increase in variability in the voltage dynamics of bursting in PD neurons pharmacologically isolated from most of the pyloric circuit (*Ratliff et al., 2021*). Can we detect similar signatures of destabilization before transitions from regular states in the intact circuit? We measured the CV of the burst periods of PD and LP neurons in regular states just before transitions away from regular (*Figure 6b*). Because we restricted our measurement of variability to regular states, we could disambiguate true cycle-to-cycle jitter in the timing of bursts from the apparent variability in cycle period due to alternations between bursting and nonbursting dynamics. We found that transitions away from regular were correlated with a steady and almost monotonic increase in variability in PD and LP burst periods for low pH and high $[K^+]$ perturbations, but not for high pH and high-temperature perturbations (Spearman rank correlation test). This suggests mechanistically different underpinnings to the pathways of destabilization between these sets of perturbations and is consistent with previous work showing that robustness to perturbations in pH only moderately affects temperature robustness in the same neuron (*Ratliff et al., 2021*).

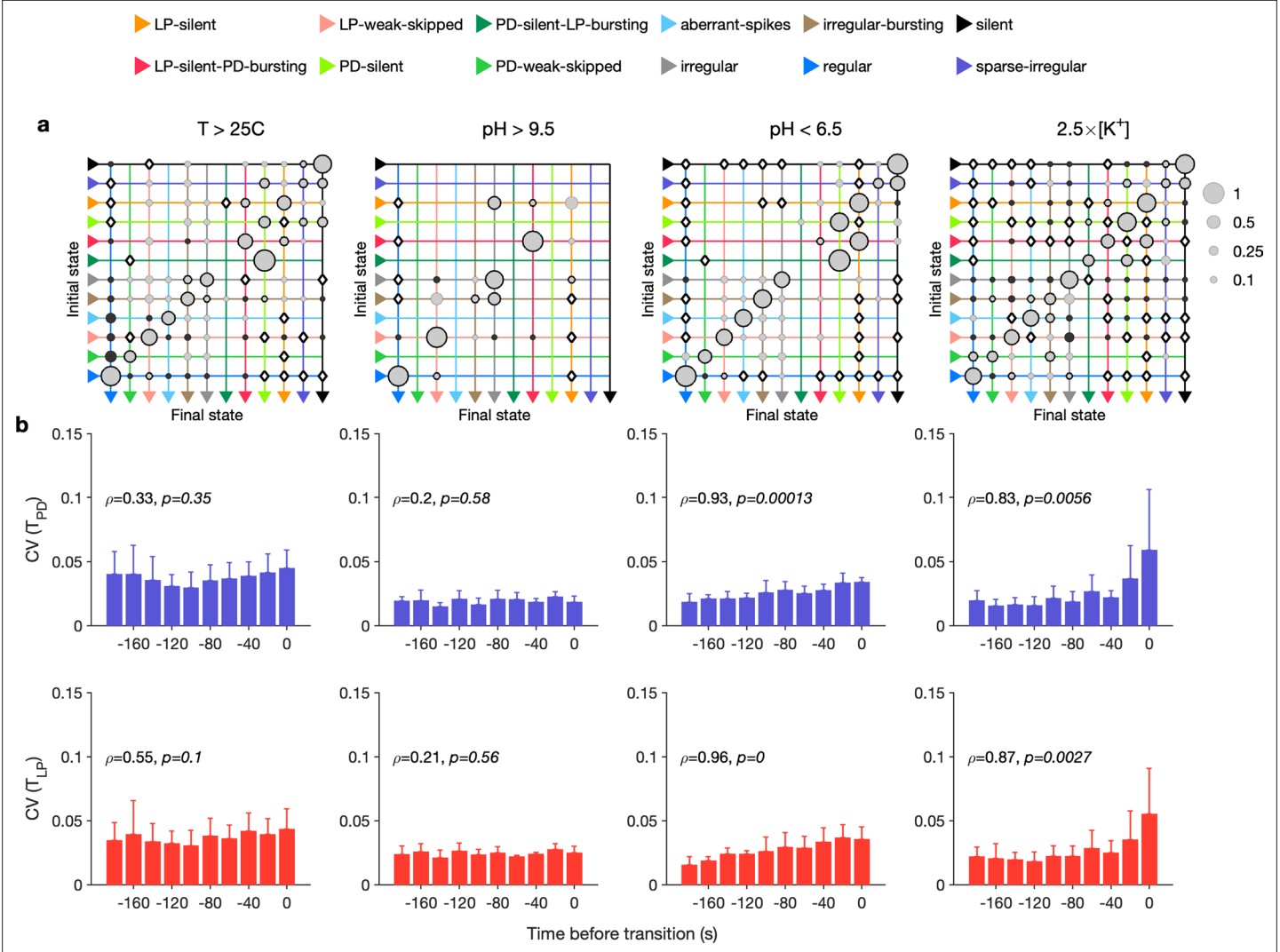

**Figure 6.** Effect of environmental perturbations on transitions between states. (**a**) Transition matrix between states during environmental perturbations. Each matrix shows the conditional probability of observing the final state in the next time step given an observation of the initial state. Probabilities in each row sum to 1. Size of disc scales with probability. Discs with dark borders are transitions that are significantly more likely than the null model (Materials and methods). Dark solid discs are transitions with nonzero probability that are significantly less likely than in the null model. ◊ are transitions that are never observed and are significantly less likely than in the null model. States are ordered from regular to silent. (**b**) Coefficient of variation (CV) of burst period of pyloric dilator (PD) (purple) and lateral pyloric (LP) (red) vs. time before transition away from the regular state. $\rho, p$ are from Spearman test (on binned data, as plotted) to check if variability increases significantly before transition. Temperature perturbations: $n = 1035$ transitions in 61 animals; pH perturbations: $n = 90$ transitions in 6 animals; $[K^+]$ perturbations: $n = 271$ transitions in 20 animals.

## Decentralization elicits variable circuit dynamics

The pyloric circuit is modulated by a large and chemically diverse family of neuromodulators that it receives via the stomatogastric (*stn*) nerve (**Marder, 2012**). Decentralization, or the removal of this neuromodulatory input via transection and/or chemical block of the *stn*, has been shown to affect the pyloric rhythm in a number of ways (**Russell, 1976**). Decentralization can stop the rhythm temporarily, which can recover after a few days (**Golowasch et al., 1999**; **Thoby-Brisson and Simmers, 1998**). Decentralization slows down the pyloric rhythm (**Eisen and Marder, 1982**; **Rosenbaum and Marder, 2018**) and makes the rhythm more variable (**Hamood and Marder, 2014**; **Hamood et al., 2015**). Decentralization can evoke variable circuit dynamics, sometimes with slow timescales (**Figure 7— figure supplement 1**), and can lead to changes in ion channel expression (**Mizrahi et al., 2001**).

The variability in circuit dynamics elicited by decentralization and the animal-to-animal variability in response to decentralization have made a quantitative analysis of the effects of decentralization

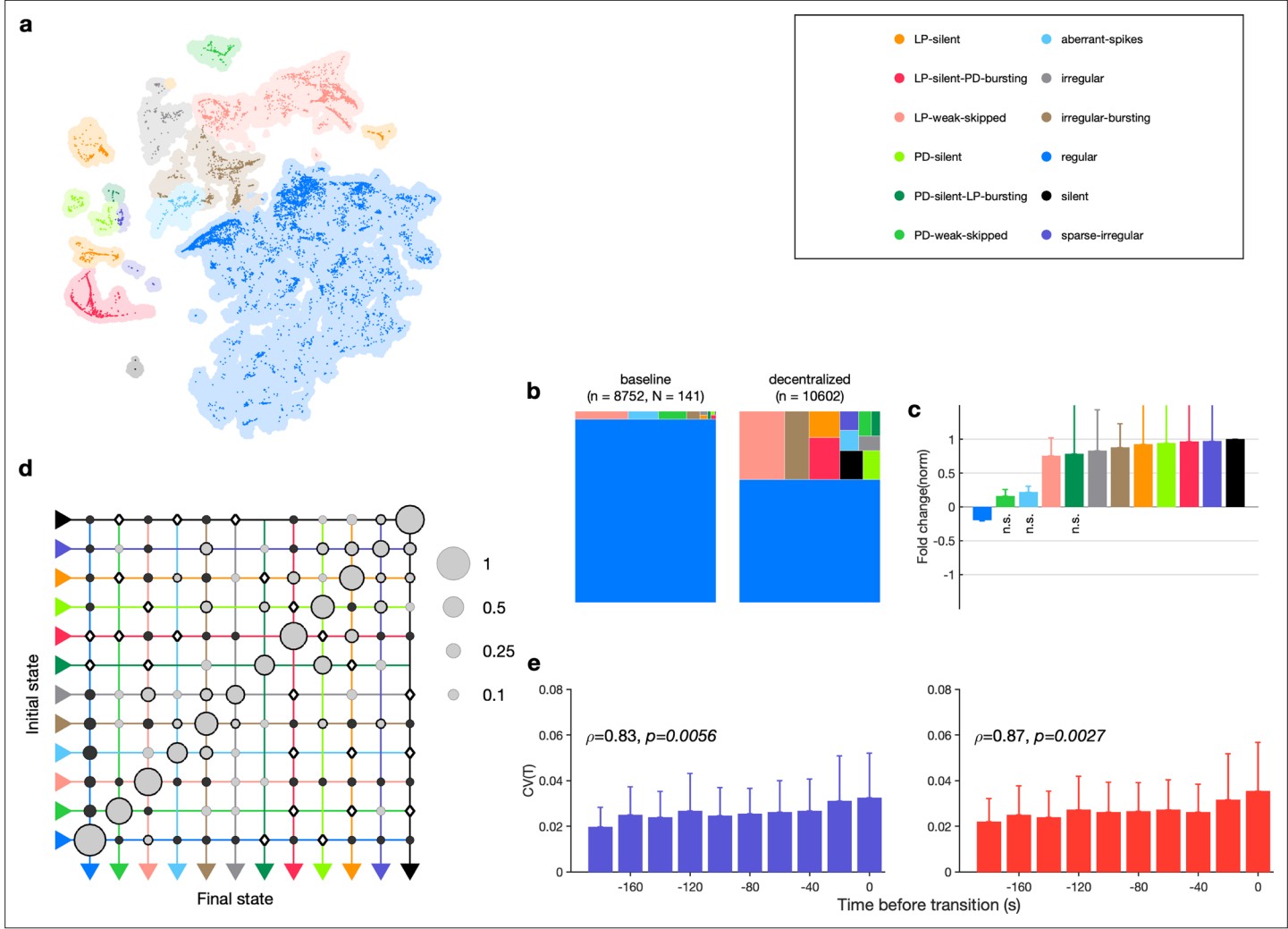

**Figure 7.** Effect of decentralization. (**a**) Map occupancy conditional on decentralization. Shading shows all data, and bright colored dots indicate data when preparations are decentralized. (**b**) State probabilities before and after decentralization. (**c**) Fold change in state probabilities on decentralization. States marked n.s. are not significantly more or less likely after decentralization. All other states are (paired permutation test, p<0.00016). (**a, b**) $n = 10,602$ points from $N = 141$ animals. (**d**) Transition matrix during decentralization. Probabilities in each row sum to 1. Size of disc scales with probability. Discs with dark borders are transitions that are significantly more likely than the null model (Materials and methods). Dark solid discs are transitions with nonzero probability that are significantly less likely than in the null model. ◇ are transitions that are never observed and are significantly less likely than in the null model. States are ordered from regular to silent. $n = 1933$ transitions. (**e**) Coefficient of variation of pyloric dilator (PD, purple) and lateral pyloric (LP; red) burst periods before transition away from regular states. $\rho, p$ from Spearman test. $n = 1332$ points from $N = 79$ animals.

The online version of this article includes the following figure supplement(s) for figure 7:

**Figure supplement 1.** Decentralization evokes variable dynamics.

**Figure supplement 2.** Effects of decentralization on state probabilities.

**Figure supplement 3.** Time course of effects of decentralization.

**Figure supplement 4.** Effects of decentralization do not correlate with seasonal effects.

difficult. We therefore set about to characterize the variable and invariant features of the changes in circuit spiking dynamics on removal of descending neuromodulation across a large ($N = 141$) population.

We first asked where in the map decentralized data were (*Figure 7a*). A large fraction ($\approx 30\%$) of the data was found outside the regular cluster, suggesting the existence of atypical circuit dynamics on decentralization. Decentralization also changed probabilities of observing many states. The regular state was significantly less likely on decentralization, and several atypical states were significantly more likely (*Figure 7b and c*, *Table 2*, *Figure 7—figure supplement 2*).

**Table 2.** State counts before and after decentralization for the data shown in *Figure 7*.
p-Values of change in probability of observing change estimated from paired permutation tests.

| State | $n_{control}$ | $n_{dec.}$ | $p$ | $\Delta P(state)$ |
|---|---|---|---|---|
| Regular | 7,967 | 5,791 | $lt_{0.001}$ | -0.308 77 |
| LP-silent | 22 | 724 | $lt_{0.001}$ | 0.030 65 |
| LP-silent-PD-bursting | 14 | 577 | $lt_{0.001}$ | 0.045 926 |
| PD-silent | 11 | 140 | 4 | 0.018 51 |
| PD-silent-LP-bursting | 20 | 18 | 0.469 59 | 0.000 188 91 |
| Aberrant-spikes | 111 | 168 | 0.300 37 | 0.003 285 3 |
| LP-weak-skipped | 317 | 1,628 | $lt_{0.001}$ | 0.099 875 |
| PD-weak-skipped | 142 | 118 | 0.292 19 | 0.003 453 8 |
| Sparse-irregular | 4 | 154 | $lt_{0.001}$ | 0.013 263 |
| Irregular | 13 | 116 | 0.000 23 | 0.010 877 |
| Silent | 0 | 321 | $lt_{0.001}$ | 0.024 825 |
| Irregular-bursting | 72 | 753 | $lt_{0.001}$ | 0.057 913 |

LP: lateral pyloric; PD: pyloric dilator.

How do preparations switch between different states when decentralized? The transition matrix during decentralization revealed many transitions between diverse states (*Figure 7d*), with the most likely transitions being significantly overrepresented compared to the null model (p<0.05, Materials and methods). Transitions away from regular included significantly more likely transitions into states where one of the neurons was irregular such as LP-weak-skipped and PD-weak-skipped. Similar to rhythm destabilization in high [$K^+$] or low pH, transitions away from regular were associated with a near-monotonic increase in the variability of PD and LP burst periods before the transitions (*Figure 7e*, $\rho \approx .8$, p<0.006, Spearman rank correlation test).

The time series of identified states on a preparation-by-preparation basis showed striking variability in the responses to decentralization (*Figure 7—figure supplement 3a*), with the probability of observing regular states decreasing immediately after decentralization (*Figure 7—figure supplement 3b*). What causes the observed animal-to-animal variability in circuit dynamics on decentralization? One possibility is that seasonal changes in environmental conditions alter the sensitivity of the pyloric circuit to neuromodulation. We tested this hypothesis by measuring the correlation between measures such as the probability of observing the regular state, the change in burst period, and the change in firing rate on decentralization and the sea surface temperature at the approximate location of these wild-caught animals (*Figure 7—figure supplement 4*). None of these measures was significantly correlated with sea surface temperature (p>0.07, Spearman rank correlation test).

## Stereotyped effects of decentralization on burst metrics

Despite the animal-to-animal variation in responses to decentralization, are there stereotyped responses to decentralization? Decentralization removes some unknown mixture of modulators that are released by the *stn*, which can vary from animal to animal. Previous work has shown that decentralization typically slows down the pyloric rhythm (*Eisen and Marder, 1982*; *Rosenbaum and Marder, 2018*) and (*Figure 8—figure supplement 1*), but a finer-grained analysis of rhythm metrics was confounded by the irregular dynamics that can arise when preparations are decentralized. For example, alteration between regular and atypical states could bias estimates of burst metrics that are not defined in atypical states. Because our analysis allows us to identify the subset of data where pyloric circuit dynamics are regular enough that burst metrics are well-defined, we measured the changes in a number of burst metrics like the burst period, duty cycle, and phases on decentralization (*Figure 8a*). Every metric measured was significantly changed except the phase at which LP bursts

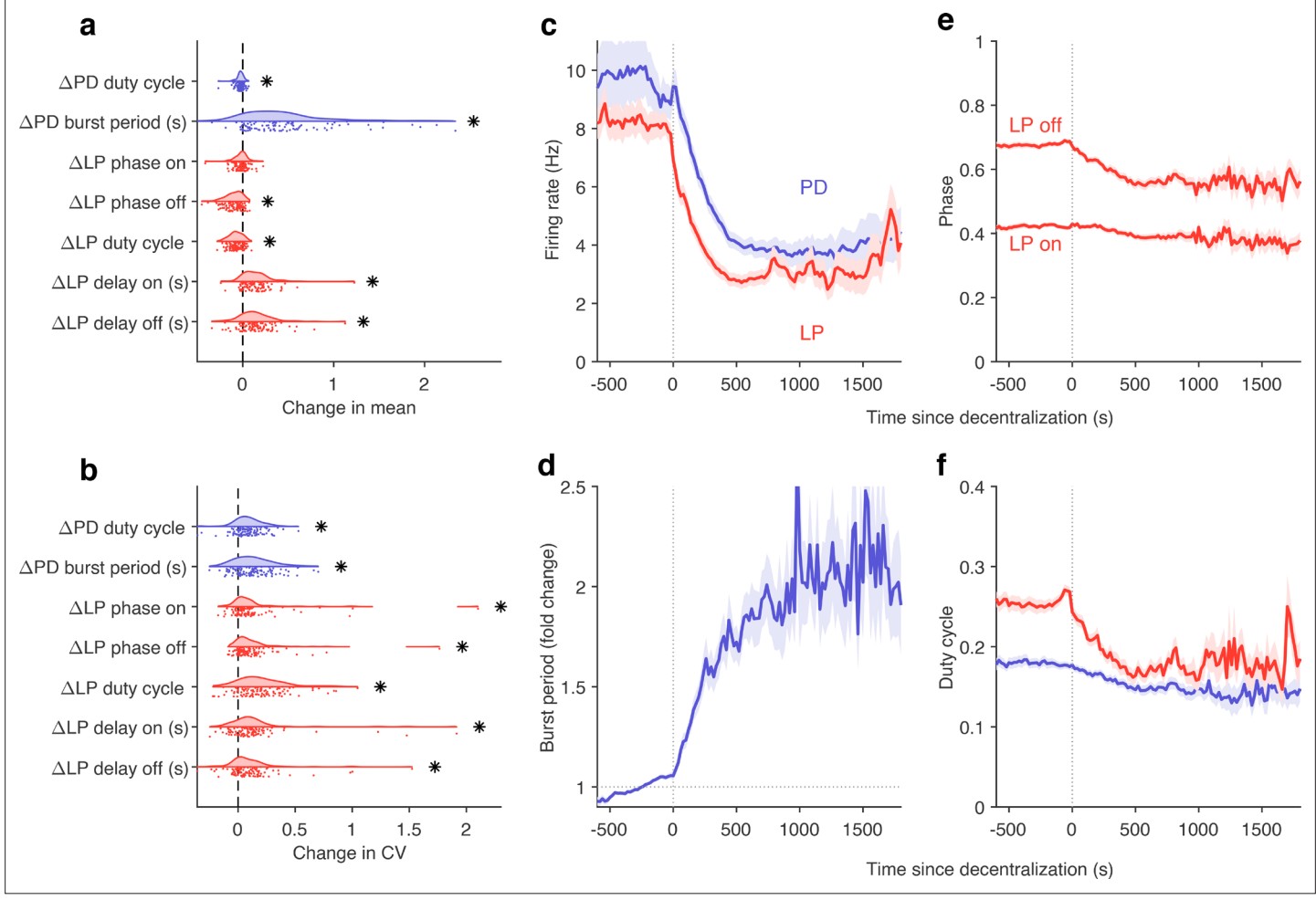

**Figure 8.** Effects of decentralization on burst metrics. (**a**) Change in mean burst metrics on decentralization. (**b**) Change in coefficient of variation of burst metrics on decentralization. In (**a**) and (**b**), each dot is a single preparation; * indicate distributions whose mean is significantly different from zero (p<0.007, paired permutation test). Firing rates (**c**), burst period (**d**), lateral pyloric (LP) phases (**e**), and duty cycles (**f**) vs. time since decentralization. In (**c–f**), thick lines indicate population means, and shading indicates the standard error of the mean. $n = 13,898$ points from $N = 141$ preparations.

The online version of this article includes the following figure supplement(s) for figure 8:

**Figure supplement 1.** Example rasters showing effect of decentralization.

**Figure supplement 2.** Effects of decentralization on regular rhythms.

start (p<0.007, paired permutation test). Consistent with earlier studies, we found that the CV in every metric increased following decentralization (*Figure 8b*).

What are the dynamics of changes in burst metrics on decentralization? Firing rates of both LP and PD neurons decreased immediately on decentralization, roughly halving their pre-decentralized values (*Figure 8c*). This occurred together with a doubling of PD burst periods (*Figure 8d*), suggesting that the entire rhythm is slowing down. Intriguingly, decentralization led to significant advance in the phase of LP burst ends, but not starts (*Figure 8e*), leading to a large decrease in the duty cycle of the LP neuron (*Figure 8f*) that was significantly more than the decrease in PD's duty cycle (p<10⁻⁸, paired *t*-test).

The stereotyped slowing of the rhythm on decentralization can also be quantified by looking at the distribution of the data in the regular cluster before and after decentralization (*Figure 8—figure supplement 2*). Data are concentrated in the upper-left edge of the regular cluster when decentralized, where burst periods are large and firing rates low (*Figure 2—figure supplement 1a and c*), suggesting that decentralization could elicit a more stereotyped rhythm for circuits that continue to burst regularly, because circuits that do so tend to share a common, slow bursting dynamics. Counterintuitively,

it may appear that regular rhythms in baseline conditions are more variable than regular rhythms after decentralization. To test this hypothesis, we measured the dispersion of each preparation in the map (*Figure 8—figure supplement 2b*) before and after decentralization. Dynamics before decentralization were significantly more dispersed in the regular cluster than dynamics after decentralization (*Figure 8—figure supplement 2c*, p = 0.0016, paired *t*-test) because they then tended to be concentrated in the upper-left edge of that cluster. To first approximation, our analysis shows that there are many ways to manifest a regular rhythm under baseline conditions, but regular rhythms on decentralization are typically slow, and stereotyped in comparison.

## Neuromodulators differentially affect state probabilities

The crustacean STG is modulated by more than 30 substances (*Harris-Warrick and Marder, 1991*; *Marder, 2012*) that tune neuronal properties at an intermediate timescale between feedback homeostasis and intrinsic cellular properties (*Daur et al., 2016*). Earlier work has focused on understanding the effect modulators have on restoring (or destabilizing) the canonical rhythm, in part because the restoration of regular oscillatory dynamics is a common feature of neuromodulator action. Other effects that neuromodulators might have on pyloric circuit dynamics are harder to investigate and are hindered by the difficulty in characterizing circuit dynamics when nonregular. Here, we set out to systematically characterize the effects of neuromodulators on dynamic states identified in the full space of circuit behaviors (*Figure 3*).

We focused our analysis on the effect of four neuromodulators: red pigment-concentrating hormone (RPCH), proctolin, oxotremorine, and serotonin. In the experiments analyzed, these neuromodulators were added to decentralized preparations so that endogenous effects of these (and other) neuromodulators were minimized. We therefore first characterized the distribution of states in decentralized preparations where neuromodulators were subsequently added (*Figure 9a*).

RPCH is a neuropeptide that targets a number of cells in the circuit (*Nusbaum and Marder, 1988*; *Swensen and Marder, 2001*) and has been shown to increase the number of spikes per burst in PD and LP (*Dickinson et al., 2001*; *Thirumalai and Marder, 2002*), though it has little effect on the pyloric period (*Thirumalai et al., 2006*). RPCH increased the probability of the regular state, suggesting stabilization of the triphasic rhythm, and decreased the probability of most other atypical states (*Figure 9b*, *Table 3*, p<0.004, paired permutation test). Consistent with earlier studies that reported that RPCH can activate rhythms in silent preparations (*Nusbaum and Marder, 1988*), the probability of observing the silent state was driven to 0 in the presence of RPCH, together with other atypical states such as LP-silent and LP-silent-PD-bursting (*Figure 9b*).

Proctolin also targets a number of cells in the circuit (*Swensen and Marder, 2001*) and strengthens the pyloric rhythm through various mechanisms: by increasing the amplitude of slow oscillations in AB and LP (*Hooper and Marder, 1987*; *Nusbaum and Marder, 1989*), depolarizing the LP neuron (*Golowasch and Marder, 1992*; *Turrigiano and Marder, 1993*), and increasing the number of spikes per burst in LP and PD (*Hooper and Marder, 1987*; *Marder et al., 1986*; *Hooper and Marder, 1984*). Oxotremorine, a muscarinic agonist, has also been shown to enhance the robustness of the pyloric rhythm (*Bal et al., 1994*; *Haddad and Marder, 2018*; *Rosenbaum and Marder, 2018*). Similar to RPCH, both proctolin and oxotremorine significantly increase the probability of the regular state (*Figure 9b*, *Table 3*, p<0.004, paired permutation test), and the regular state is the only one significantly more likely when the neuromodulator is added. The strengthening effects of RPCH and oxotremorine are also manifested in the significantly lower probabilities of observing atypical and dysfunctional states such as silent, LP-silent, PD-silent, and sparse-irregular (*Table 3*).

Serotonin can have variable effects on the pyloric circuit, varying from animal to animal, and can either speed up or slow down the rhythm (*Beltz et al., 1984*; *Spitzer et al., 2008*). In *Panularis*, serotonin depolarizes LP in culture, but hyperpolarizes LP in situ, unlike other neuromodulators that typically have the same effect in situ and in culture (*Turrigiano and Marder, 1993*). Consistent with earlier work in *C. borealis* showing that serotonin destabilizes the rhythm in decentralized preparations (*Haddad and Marder, 2018*), we found that the probability of regular states was significantly lower on addition of serotonin (*Figure 9b*, *Table 3*, p<0.004, paired permutation test), together with a significantly higher probability of atypical dysfunctional states such as LP-silent, aberrant-spikes, PD-silent-LP-bursting, and irregular, suggesting loss of coordination between the many neurons in the pyloric circuit with serotonin receptors (*Clark et al., 2004*).

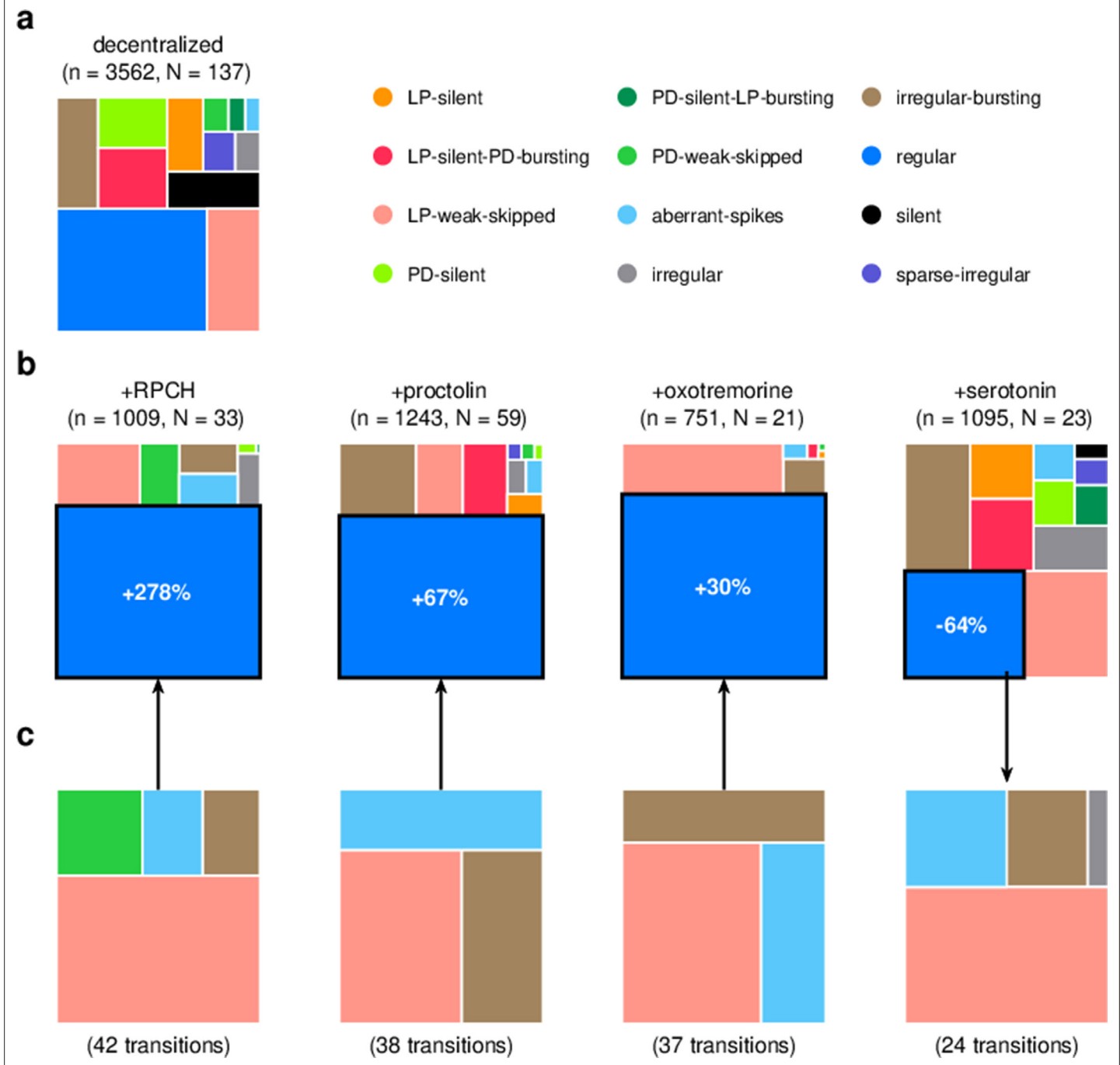

**Figure 9.** Effect of bath-applied modulators. (**a**) State distribution in decentralized preparations. (**b**) State distribution in bath application of neuromodulators. Change percentages show difference in probability of regular state from decentralized to addition of neuromodulator. (**c**) Probability distribution of states conditional on transition to (for red pigment-concentrating hormone [RPCH], proctolin, and oxotremorine) or from (for serotonin) the regular state. $n$ is the number of data points, and $N$ is the number of animals.

The online version of this article includes the following figure supplement(s) for figure 9:

**Figure supplement 1.** Raw traces during proctolin application.

**Figure supplement 2.** Neuromodulators affect map occupancy.

**Table 3.** Probability distribution of states during modulator application, as shown in *Figure 9*.

| State | Decentralized | RPCH | Proctolin | Oxotremorine | Serotonin |
|---|---|---|---|---|---|
| Regular | 0.39 | 0.73 | 0.69 | 0.78 | 0.27 |
| LP-silent | 0.06 | 0 | 0.02 | 0 | 0.07 |
| LP-silent-PD-bursting | 0.09 | 0 | 0.07 | 0 | 0.1 |
| PD-silent | 0.07 | 0 | 0 | 0 | 0.04 |
| PD-silent-LP-bursting | 0.01 | 0 | 0 | 0 | 0.03 |
| Aberrant-spikes | 0.01 | 0.04 | 0.01 | 0.01 | 0.03 |
| LP-weak-skipped | 0.14 | 0.11 | 0.07 | 0.17 | 0.19 |
| PD-weak-skipped | 0.02 | 0.05 | 0 | 0 | 0 |
| Sparse-irregular | 0.03 | 0 | 0.01 | 0 | 0.02 |
| Irregular | 0.02 | 0.02 | 0.01 | 0 | 0.07 |
| Silent | 0.07 | 0 | 0 | 0 | 0.01 |
| Irregular-bursting | 0.1 | 0.04 | 0.11 | 0.03 | 0.17 |

LP: lateral pyloric; PD: pyloric dilator.

Do these modulators share common features in how they (de)stabilize the rhythm? We computed the probability distribution of states conditional on transitions to the regular state for RPCH, proctolin, and oxotremorine, and conditional on transitions from the regular state for serotonin (*Figure 9c*). For all four neuromodulators, the conditional state distribution predominantly comprised these three states: LP-weak-skipped, irregular-bursting, and aberrant-spikes, suggesting that trajectories of recovery or destabilization of the regular rhythm share common features. Serotonin destabilizes the rhythm, decreasing the likelihood of observing regular states, similar to environmental perturbations (*Figure 5*) and decentralization (*Figure 7*).

Different neuromodulators activate different forms of the rhythm (*Marder and Weimann, 1992*; *Marder et al., 1985*; *Marder, 2012*), partly because different neuron types express different receptors to varying extents (*Garcia et al., 2015*). Moreover, similar rhythmic motor patterns can be produced by qualitatively different mechanisms, such as one that depends on voltage-gated sodium channel activity, and one that can persist in their absence (*Harris-Warrick and Flamm, 1987*; *Epstein and Marder, 1990*; *Rosenbaum and Marder, 2018*). To determine if different neuromodulators elicit regular rhythms that occupy different parts of the map, we plotted the location of data elicited by various neuromodulators in the full map (*Figure 9—figure supplement 2*). Regular data elicited by different neuromodulators tended to lie in clusters, whose distribution in the map was significantly different between serotonin and CCAP (Crustacean cardioactive peptide), and proctolin and every other neuromodulator tested ($p < 0.05$, two-dimensional Kolmogorov–Smirnov test, using the method of *Peacock, 1983*). The differential clustering of regular states in the map with neuromodulator suggests that neuromodulators can elicit characteristic, distinct rhythms.

## Neuromodulators differentially affect transition between states

RPCH, proctolin, and oxotremorine activate a common voltage-dependent modulatory current, $I_{MI}$ (*Swensen and Marder, 2001*), but can differentially affect neurons in the STG because different cell types express receptors to these modulators to different degrees. For example, RPCH activates $I_{MI}$ strongly in LP neurons, but the effects of oxotremorine and proctolin are more broadly observed in the circuit (*Swensen and Marder, 2000*; *Swensen and Marder, 2001*). Though these three modulators strengthen the slow-wave oscillations in pyloric neurons, only oscillations elicited by oxotremorine and RPCH persist in tetrodotoxin, and proctolin rhythms do not, hinting that qualitatively different mechanisms underlie the generation of these seemingly similar oscillations (*Rosenbaum and Marder, 2018*). We therefore measured the transition rates between states during neuromodulator application to how similar or different trajectories towards recovery were.

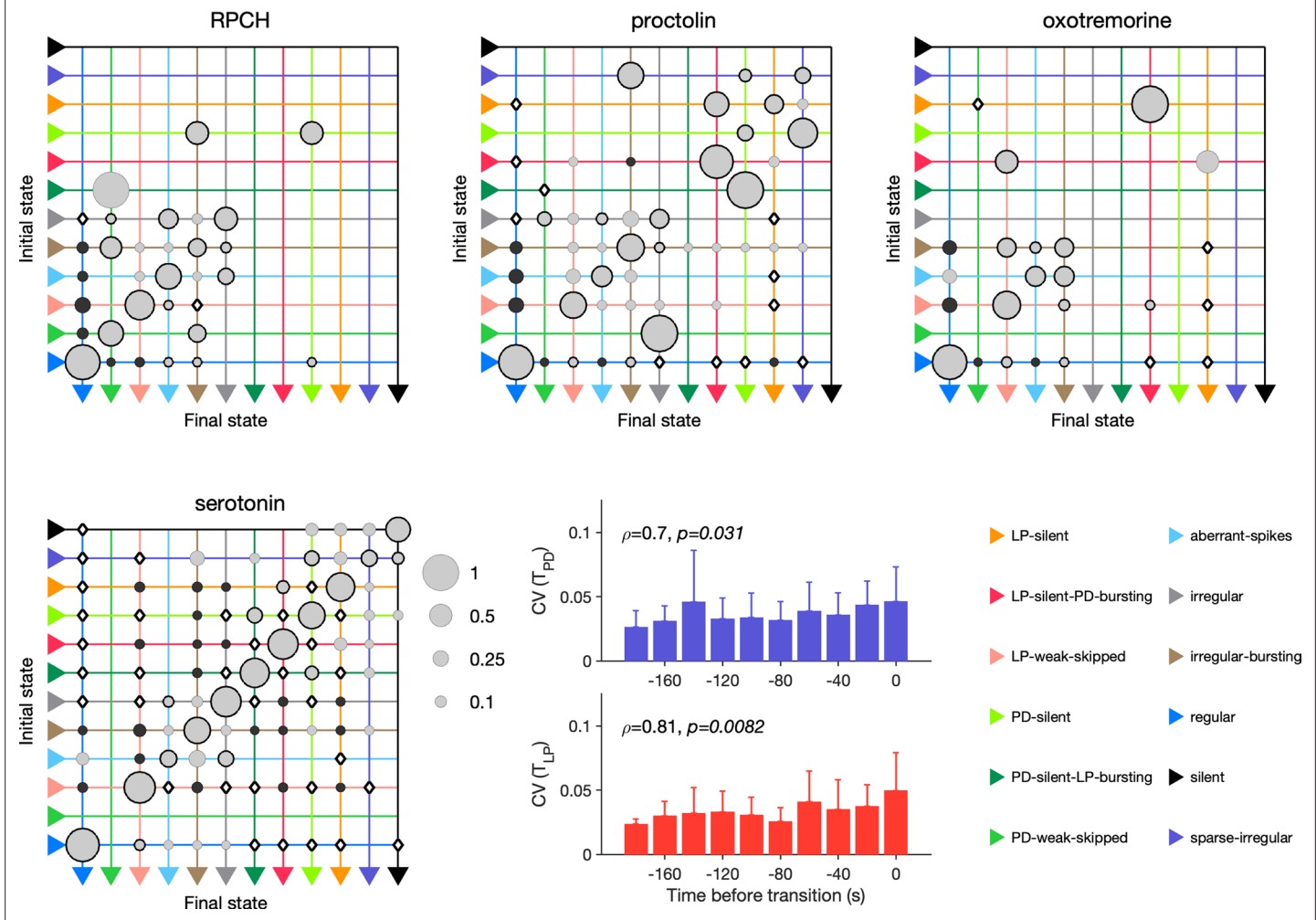

**Figure 10.** Effect of red pigment-concentrating hormone (RPCH), proctolin, oxotremorine, and serotonin on transition probabilities. Each matrix shows the conditional probability of observing the final state in the next time step given an observation of the initial state during bath application of that neuromodulator. Probabilities in each row sum to 1. Size of disc scales with probability. Discs with dark borders are transitions that are significantly more likely than the null model (Materials and methods). Dark solid discs are transitions with nonzero probability that are significantly less likely than in the null model. ◇ are transitions that are never observed and are significantly less likely than in the null model. States are ordered from regular to silent. Bar graphics show the coefficient of variability (CV) of pyloric dilator (PD) and lateral pyloric (LP) burst periods before transition away from regular states. $\rho, p$ from Spearman rank correlation test. RPCH: $n = 148$ transitions in $N = 33$ animals; proctolin: $n = 155$ transitions in $N = 59$ animals; oxotremorine: $n = 102$ transitions in $N = 21$ animals; serotonin: $n = 263$ transitions in $N = 23$ animals. Bar graphs show the CV of burst periods of PD and LP vs. time before a transition away from regular states during serotonin application. $\rho, p$ from Spearman rank correlation test.

In RPCH, proctolin, and oxotremorine application, ≈100 transitions were observed between states (*Figure 10*). Transitions could not always be predicted by a null model assuming that transition probabilities scaled with the conditional probability of observing states after a transition. For example, some transitions, such as the transition from irregular to regular, were never observed in RPCH, a significant deviation from the expected number of transitions given the likelihood of observing regular states after transitions (Materials and methods). Others, such as the transition LP-silent to LP-silent-PD-bursting in proctolin and oxotremorine, were observed at rates significantly higher than expected from the null model. Transitions into regular state are distributed across aberrant-spikes, LP-weak-skipped, and irregular-bursting states for all three, but no invariant feature emerges in the rest of the transition matrix.

Serotonin destabilizes the rhythm in decentralized preparations, and the transition matrix under serotonin reveals several features of the irregularity behavior observed under serotonin (*Figure 10*). A number of irregular and low-firing states from silent to irregular never transition into the regular

state, which is unlikely in the null model (p<0.05, Materials and methods). Transitions between pairs of states are symmetric and occur at rates significantly larger than in the null model, such as between LP-silent and LP-silent-PD-bursting. Intriguingly, destabilizing transitions from regular to LP-weak-skipped, aberrant-spikes, and irregular-bursting are observed at rates significantly higher than in the null model. These three abnormal states are also observed immediately preceding regular states in RPCH, proctolin, and oxotremorine (*Figure 9c*), suggesting that the mechanisms for both stabilization and destabilization of the rhythm share stereotyped trajectories.

Are transitions away from regular states also associated with increases in variability of burst periods? Similar to preparations in high $[K^+]$ and low pH, and when decentralized, transitions away from regular states in serotonin were associated with significantly rising variability in the burst periods of PD and LP neurons (*Figure 10*, p<0.05, Spearman rank correlation test).

## Discussion

Advances in neural recording technology have made it possible to generate increasingly large data-sets, and an ongoing challenge is in developing computational tools to find structure in the neural haystack (*Pachitariu et al., 2016*). Nonlinear dimensionality reduction algorithms such as t-SNE can create a useful representation of datasets that are too large to visualize in their entirety using traditional methods. We combined domain-specific expert knowledge with an unsupervised dimen-sionality reduction process (t-SNE) by manually segmenting and labeling clusters of dynamics repre-senting biologically significant behavior. This approach conferred two advantages: it allowed for a more accurate measure of traditional metrics such as burst phases in large datasets (*Figures 4 and 8*), and it allowed for the analysis of irregular dynamics that are typically intractable with conventional analysis methods (e.g., *Figure 9*), with the disadvantage of not being fully automated, and requiring human intervention to inspect data in the embedding and draw cluster boundaries. Our work hints at a possibility to characterize nonregular spike patterns in small neural circuits and can thus provide a deeper understanding of circuit activity under baseline conditions and in response to perturbations. Our approach makes limited assumptions of the dynamics of the circuit, yet provides a formal frame-work based on domain-specific knowledge for characterizing circuit activity. Additionally, this way of analyzing neural spike data can readily be adapted to other circuits and systems.

### Reliable identification of regular rhythms allows for accurate, interpretable analysis of rhythm metrics

Characterizing the statistics of neural oscillations has several subtle challenges. For example, varia-tions in cycle period arising from cycle-to-cycle fluctuations are not distinguished from those arising from alteration between epochs of regular oscillations interrupted by spans of irregular activity where metrics like cycle period are undefined. One way to disambiguate the two is to construct elaborate checks to make sure that the spike pattern being measured meets certain criteria. However, edge cases abound, and this is a challenging and subjective approach. A fortuitous consequence of the embedding method we used is to reliably identify when rhythms were regular, and we found that burst metrics were well defined for this subset of data (blue region in *Figure 3*). We were therefore able to measure the mean and variability of various burst metrics (*Figure 4*) only in stretches of data where it made sense to do so, and thus the measured variability stemmed almost entirely from cycle-to-cycle variations.

Consistent with previous studies (*Bucher et al., 2005*; *Hamood and Marder, 2014*; *Hamood et al., 2015*), our results (*Figure 4*) show that within-animal variability in pyloric burst metrics is lower than across-animal variability. Our results are from an analysis of data from several experimenters from different laboratories, collected over a span of 10 years. It is therefore an ideal dataset in which to measure variability. We find that the CV of all burst metrics measured is $\approx 0.1$ (*Figure 4b*), which can now be used as a standard for regular baseline pyloric oscillations. Measuring burst metrics on decen-tralization (*Figure 8*) also allowed us to characterize how regular rhythms change, while still being recognizably regular. In addition to recapitulating well-understood phenomena such as the slowing down and increased variability in rhythms, we found that the phase of LP burst onset did not change significantly, but the phase of LP burst termination did, suggesting that features of the rhythm are differentially robust to the removal of neuromodulation.

Earlier work categorized the varied dynamics of the pyloric circuit during perturbations (*Haddad and Marder, 2018*; *Haley et al., 2018*; *Ratliff et al., 2021*). In those studies, categories were typically constructed by hand and were not rigorously shown to be mutually exclusive. Categories in this work, while manually chosen, emerge naturally from the distribution of the data in the reduced space (*Figure 3*) and no segment of data can have more than one label because it can exist only at a single point in the map. For instance, earlier work categorized rhythms that were labeled regular into two categories, 'normal triphasic' and 'normal triphasic slow' (*Haddad and Marder, 2018*), while we did not observe a distinctly bimodal distribution of burst periods. In contrast, the catch-all 'atypical firing' state was separated here into a number of states (irregular, irregular-bursting, sparse-irregular) that span several well-separated clusters in the map (*Figure 3*). In summary, this work recapitulates every label constructed to categorize spike patterns from PD and LP neurons in earlier work, and additionally finds new spike patterns that were either not detected or not identified as distinct because they are hard to detect by manual inspection.

## Diversity and stereotypy in trajectories from functional to crash states

Are there preferred paths to go from regular rhythms to crash? Diversity in the solution space of functional circuits, and the varied effects of perturbations on these circuits, argues for an assortment of trajectories from functional dynamics to irregular or silent states. While transition matrices measured during different perturbations were varied (*Figure 6*), we did observe universal features in transition matrices measured during environmental perturbations, decentralization, and addition of neuromodulators (*Figures 6, 7 and 10*). The destabilizing transition from regular → LP-weak-skipped was overrepresented in every transition matrix, suggesting that the weakening of the LP neuron is a crucial step in the trajectories towards destabilization, perhaps because there is only one copy of LP in the circuit. Earlier work studying trajectories of destabilization of regular bursting in the isolated pacemaker kernel also found a conserved motif in trajectories towards destabilization: from regular bursting to tonic spiking to silence in response to pH perturbations, and another conserved motif (bursting to weak bursting to silence) in response to temperature perturbations (*Ratliff et al., 2021*). Transitions away from regular rhythms were also associated with increased variability in burst periods during all perturbations except high temperature and low pH (*Figures 6, 7 and 10*). An increase in variability in PD voltage dynamics before transitions from regular bursting has been observed in the isolated pacemaker kernel (*Ratliff et al., 2021*), similar to the effect we observed in the intact circuit.

The structure of the transitions between states also hints at features of the circuit that are critical for rhythm (de)stabilization. Unsurprisingly, PD-silent states precede silent states in low pH, high temperature, and high $[K^+]$ perturbations (*Figure 6*). This makes sense because PD cells are electrically coupled to the endogenous burster AB in the pacemaker kernel, and silencing the pacemaker kernel can cause the circuit to go silent. Though the states are determined purely from clusters in the embedding (*Figure 2*), and thus from statistical features of spike times, some states may be identified predominantly with cell-specific features (e.g., LP-weak-skipped where the LP neuron fails to burst regularly, but the PD neurons continue to burst regularly), or with circuit-level features (e.g., aberrant-spikes where one or both neurons fire spikes outside the main burst, which may be caused by incomplete inhibition). Decentralization elicits the largest number of transition types, with ≈ 80% of all transition types observed, which could be a consequence of the complex change in the neuromodulator milieu following transection of descending nerves.

## Linking circuit output to circuit mechanisms

A large body of work has shown that there is more than one way to make a neural circuit with similar patterns of activity (*Prinz et al., 2003*; *Prinz et al., 2004*; *Gutierrez et al., 2013*). Several combinations of circuit parameters such as synapse strengths, ion channel conductances, and network topology can be found in circuits that generate similar emergent collective dynamics (*Gonçalves et al., 2020*). The dimensionality of the space of neuronal and synaptic parameters in a neural circuit is much larger than the dimensionality of the circuit output (*Marder and Bucher, 2007*). This disparity in dimensionality leads to an inherently many-to-one mapping from the space of circuit architecture to the space of circuit dynamics. Circuits can therefore exhibit 'cryptic' architectural variability (*Haddad and Marder, 2018*), where the diversity of topologies and neuronal parameters is masked by the relatively low-dimensional nature of the observed circuit outputs. However, perturbations can reveal differences

between seemingly identical circuits. For instance, current injections in an oscillator network can shift phases, thus revealing connection weights between individual neurons (*Timme, 2007*). This work reveals a path towards analysis that can reveal cryptic variability and build mechanistic links from circuit architecture to function. By characterizing the totality of circuit dynamics under a variety of conditions, our framework provides a way to fit biophysically detailed models of the pyloric circuit to diverse circuit dynamics under baseline conditions and perturbations. From the large diversity of neuron and circuit parameters that can reproduce a snapshot of activity, only a subset of models could potentially recapitulate the diverse irregular behavior seen under extreme perturbations. Recent work that reproduced how circuits change cycle periods with temperature (*Alonso and Marder, 2020*) can be extended to find parameter sets that also generate the irregular states characterized in this study at the rates observed in the data. Crucially, the characterization of the pyloric circuit dynamics in this work can be used to rule out models and parameter sets that generate irregular activity that is qualitatively dissimilar to any of the irregular states observed in the pyloric circuit. Future experimental work can pair data analysis methods such as this work with quantitative measurements of cellular and circuit parameters using emerging techniques (*Schulz et al., 2006*; *Schulz et al., 2007*; *Tobin et al., 2009*) to find parameter sets that generate robust rhythms and irregular states.

## Applicability to other systems

The analysis method in this study is well-suited for large datasets of neural recordings from identified neurons. Data where the identity of each neuron is not or cannot be known, such as large-scale mammalian brain recordings, would require modifications to the analysis pipeline described in *Figure 2*. First, it would no longer be possible to construct a data vector of fixed length because ordering of the different neurons would not be meaningful. Each data point would instead be an unordered set of spike times from each neuron, and a distance function that operated on spike times (*Christen et al., 2006*; *Victor and Purpura, 2009*; *Schreiber et al., 2003*; *van Rossum, 2001*) could be used to generate a distance matrix between raw data points, which would be the input to the embedding algorithm. In our analysis, we included features such as the 'spike phase' (*Figure 2b and c*) because the neurons in this circuit interact with one another strongly in each cycle of oscillations. The analysis of neural circuits that do not show such strong intrinsically phase-controlled behavior could use other features more suitable to those systems.

## Comparison with other methods

Visualization and other forms of analysis of large neural datasets rely on dimensionality reduction (*Nguyen and Holmes, 2019*). Here, we used the t-SNE algorithm as a core method to reduce the dimensionality of the dataset and visualize our data. t-SNE has been widely used in the unsupervised analysis of many types of biological data (*Berman et al., 2014*; *Kollmorgen et al., 2020*; *Chen et al., 2021*; *Macosko et al., 2015*; *Kobak and Berens, 2019*; *Leelatian et al., 2020*), including neural recordings (*Dimitriadis et al., 2018*). t-SNE is a technique that allows high-dimensional data to be visualized in a lower-dimensional space (*Van der Maaten and Hinton, 2008*; *Linderman and Steinerberger, 2019*), and works by preserving pairwise distances between points in the high-dimensional space and the low-dimensional embedding, within a certain neighborhood. This feature makes t-SNE an attractive tool to try to visualize large, structured datasets, such as those examined in this study, because it can demonstrate how similar spike patterns are to each other (*Dimitriadis et al., 2018*). t-SNE has been shown rigorously to be capable of recovering well-separated data clusters (*Linderman and Steinerberger, 2019*). In our application, t-SNE generated embeddings where spike patterns in different regions could be described as qualitatively different. For example, spike patterns in the topmost cluster (colored green in *Figure 3*) all had weak PD spiking, but regular and strong LP spiking. This was qualitatively different from the two closest clusters LP-weak-skipped and irregular. In regions of the map where clusters were not cleanly separated (e.g., in the connection between the regular and irregular-bursting clusters), manual inspection revealed a number of intermediate states. The clustered or not-clustered regions of the map are therefore informative of the underlying distribution of spike patterns and emerge robustly from the embedding.

t-SNE is widely used in the analysis and visualization of high-dimensional data, but is important to acknowledge its limitations. t-SNE can generate embeddings that appear to have clusters from purely randomly distributed data, can distort sizes of clusters, and can fail to preserve large-scale topological

features of the data in some embeddings (*Wattenberg et al., 2016*). The visualization we generated was useful in that it guided manual clustering and made feasible a previously intractable task, that of classifying hundreds of hours of spike patterns from hundreds of animals.

A variety of other dimensional reduction techniques, including multidimensional scaling (*Cox and Cox, 2008*), convolutional non-negative matrix factorization (*Mackevicius et al., 2019*) and their extensions (*Williams et al., 2020*), tensor component analysis (*Williams et al., 2018*), and dynamical component analysis (*Clark et al., 2019*), have been developed that aid in visualizing and analysis of large neural datasets. Methods based on neural networks offer powerful tools to analyze unstructured neural data by modeling the data with a recurrent neural net and then analyzing that model (*Vyas et al., 2020*). Topological autoencoders are one such technique that combine autoencoders with methods from topological data analysis to produce representation in lower-dimensional spaces (*Moor et al., 2019*). These methods are similar in spirit to the analysis presented here, but use sophisticated neural nets whose parameters yield the lower-dimensional representation. Other analysis methods include SOM-VAE, which combines self-organizing maps (SOMs) and variational auto-encoders (VAEs) (*Fortuin et al., 2018*) to analyze high-dimensional time series and find transitions between states, and deep temporal clustering, which combines dimensionality reduction and temporal clustering (*Madiraju et al., 2018*).

## Technical considerations

In this study, we have used the activity of the LP and PD neurons as a proxy for the pyloric circuit. However, the pyloric circuit contains other neurons: AB (which is electrically coupled to PD neurons), PY neurons (which are anti-phase to both PD and LP), and VD and IC neurons. A richer description of the dynamics of the pyloric circuit would include spikes from these neurons, and the methods we have described here can be scaled up to include these neurons. It is likely that we are underestimating the number of states, and thus, transitions between states, because we do not have access to the dynamics of these neurons. Datasets that contain recordings from all pyloric neurons as preparations are subjected to the perturbations studied here and are not available for large numbers of animals. We therefore chose to focus on the functional antagonists LP and PD. Additionally, neurons in the pyloric circuit are coupled using graded synapses, and the circuit can generate coordinated activity even when spiking is abolished (*Rosenbaum and Marder, 2018*), suggesting that subthreshold oscillations may be an important feature we are not measuring by only recording spikes. However, the data required necessitates substantially harder to perform experiments because intracellular electrodes must be used. Furthermore, the signal to the muscles – arguably the physiologically and functionally relevant signal – is the spike signal, suggesting that spike patterns from the pyloric circuit are a useful feature to measure.

The unit of data we operated on was a time series of spikes from the LP and PD neurons. In order to describe what the dynamics of these neurons is at a given point in time, we chose to look at a neighborhood in time. In this article, we chose 20 s nonoverlapping bins, based on inspection of the data by eye. Choosing a time bin imposes certain tradeoffs in the analysis of time series: changes in dynamics on timescales smaller than the bin are counted as different states, and changes in dynamics on timescales longer than the bin size are counted as transitions between states. The statistics of the transitions we measure are therefore dependent on the bin size we chose. We note that dwell times in each state are almost always in excess of null model predictions generated by shuffling states (transition matrices in *Figures 6, 7 and 10*), supporting the validity of our choice of 20 s bins.

## Conclusion and outlook

Our work provides a way to characterize nonregular spike patterns in small neural circuits. It thus provides a bridge between experimental or simulation work grounded in the biophysical detail of ion channels and synaptic currents; and the rich body of observations of circuits under baseline and perturbed conditions. The methods we have employed can easily be adapted to other circuits and systems, make limited assumptions of the dynamics of the circuit, yet provide a robust framework on which to hang a large volume of previously ineffable expert domain knowledge.

Prior to this work, crashes in the pyloric circuit and irregular dynamics in a normally regular circuit were difficult to characterize. We present a method to tame the complexity of the distribution of irregular states by exploiting the fact that pyloric dynamics are not unbounded even in their irregularity. By

using a t-SNE in conjunction with manual inspection of reduced data and manual clustering, we have made this previously intractable problem feasible and found undiscovered spike patterns and transitions. Our approach was successful because we used a dataset with long recordings from identified neurons in a circuit that can be subjected to many different perturbations, which is one of the advantages of using the STG system. It will be interesting to see if this can be applied to other systems with identified neurons in a functional circuit to characterize their function and failure modes.

In intact pyloric circuits, and in the presence of modulatory input from the *stn*, almost all networks are 'normal' and exhibit regular rhythms. Decentralization can generate more variable dynamics, presumably because the underlying differences in network structure that were compensated by modulator action now manifest as different collective dynamics in the network. Although it may appear that modulators can have similar effects when added to a decentralized network, they are in fact distinguishable when looking at how they influence the totality of circuit dynamics, not just the regular state.

A major unanswered question was whether crashes triggered by different perturbations share dynamical mechanisms and common pathways. Earlier work looking at a simpler subset of the pyloric circuit argued that different perturbations led to stereotyped but diverse transitions before crash, and we have extended this result in the intact circuit. We show that different perturbations can have different trajectories to crash, but the stereotypy observed in the simpler system was not observed, presumably due to the larger number of pathways accessible to the intact circuit. The new insight from this work stems from the fact that this is the first time transitions through multiple physiological conditions in so many modalities have been characterized and shows that there are many paths through possible circuit dynamical states from canonical states to crash. Several studies focus on one perturbation at a time. By studying a number of perturbations together, we compare responses to different kinds of perturbations on the same physiological network.

## Materials and methods
### Animals and experimental methods
Adult male Jonah crabs (*C. borealis*) were obtained from Commercial Lobster (Boston, MA), Seabra's Market (Newark, NJ), and Garden Farm Market (Newark, NJ). Dissections were carried out as previously described (*Gutierrez and Grashow, 2009*). Decentralization was carried out either by cutting the *stn* or by additionally constructing a well on the *stn* and adding sucrose and TTX (tetrodotoxin) as described in *Haddad and Marder, 2018*. Temperature was controlled as described in *Tang et al., 2010*; *Tang et al., 2012*; *Haddad and Marder, 2018*. Extracellular potassium concentrations were varied as described in *He et al., 2020*. pH perturbations are described in *Haley et al., 2018*.

### Data selection and curation
Our goal was to include as much data as possible to create as complete a description of pyloric dynamics as possible. Following our strategy of including only the LP and PD neurons, we used every available dataset that recorded from these neurons from the Marder lab. We also included available datasets from the Nadim and Bucher labs. No dataset was explicitly excluded for reasons linked to the activity of the pyloric circuit in those datasets. Data where crucial metadata was not recorded (e.g., if the temperature of the preparation was not recorded) was excluded. Data where only *lvn* was recorded from was only included in cases of exceptional data quality, where it was judged that PD and LP could be reliably identified.

### Spike identification and sorting
Spikes are identified from extracellular recordings of motor nerves or from intracellular recordings. LP spikes were identified from intracellular recordings, *lvn*, *lpn,* and *gpn* nerves (in descending order of likelihood). PD spikes were identified from *pdn*, intracellular recordings, and *lvn*. We used a custom-designed spike identification and sorting software (called 'crabsort') that we have made freely available at https://github.com/sg-s/crabsort (copy archived at swh:1:rev:6a67e765e90caa536e6a11f67d9d4737d059af50; *Gorur-Shandilya, 2021*), previously described in *Powell et al., 2021*. Spikes are identified using a fully connected neural network that learns spike shapes from small labeled datasets. A new network is typically initialized for every preparation. Predictions from the neural network also indicate the confidence of the network in these predictions, and uncertain predictions are inspected

and labeled and the neural network learns from these using an active learning framework (**Settles, 2009**).

## Data curation and data model

Each file was split into 20 s nonoverlapping bins, and spike times, together with metadata, were assembled into a single immutable instance of a custom-built class (embedding.DataStore). The data store had the following attributes:

- *Spike times* containing LP and PD spike times.
- *ISIs* containing ISIs and spike phases
- *Labels* categorical data containing manually generated labels from **Figure 3**
- *Metadata* such as concentration of modulators, pH, temperature, whether the preparation was decentralized or not, etc.

Using an immutable data structure, reduced risks of accidental data alteration during analysis. Every attribute was defined for every data point.

## Embedding

### ISI and phase representation (Figure 2b)

Each data point is a 20 s bin containing spike times from LP and PD neurons (**Figure 2a**). For each data point, spike times are converted into ISIs. A set of spike times uniquely identifies a set of (ordered) ISIs. The set of LP spike times generates a set of LP ISIs, and the set of PD spike times generates a set of PD ISIs (**Figure 2b**).

For every spike in PD or LP, a 'spike phase' can be calculated as follows. Spike phases are not defined when either LP or PD are silent in that data point, or for LP/PD spikes with no spikes from the other neuron before or after that spike. Thus, the 'spike phase' of the $i$th spike on neuron X w.r.t. neuron Y is given by

$$\frac{t_i^X - t_{i,-}^Y}{t_{i,+}^Y - t_{i,-}^Y} \in [0, 1]$$

where $t_i^X$ is the time of the $i$th spike on neuron $X$, $t_{i,-}^Y$ is the time of the last spike on $Y$ before $t_i^X$, and $t_{i,+}^Y$ is the time of the first spike after $t_i^X$. Note that this definition can be generalized to $N$ neurons, though the number of spike phases grows combinatorially with $N$.

### Construction of vectorized data frame (Figure 2c and d)

Each data point can contain an arbitrary number of spikes, and thus an arbitrary number of ISIs and spike phases. Ideally, each data point is a data frame of fixed length (a point in some fixed high-dimensional space). To do so, we computed percentiles ISIs and spike phases (**Figure 2c**). We chose 10 bins per ISI type (deciles). The end result is not strongly dependent on the number of bins chosen as long as there are sufficiently many bins to capture the distinctly bimodal distribution in ISIs during bursting.

We included four other features to help separate spike patterns that appeared qualitatively different. First, firing rates of LP and PD neurons. Second, the ratios of second-order to first-order ISIs, defined as

$$\frac{\max I^{(2)}}{\max I^{(1)}}$$

where $I^{(n)}$ is the nth order set of ISIs computed as the time between n spikes. $I^{(1)}$ is the simple set of ISIs defined between subsequent spikes. This measure is included because it captures the difference between single spike bursts and normal bursts well. Third, the ratio between the largest and second-largest ISIs for each neuron.

Finally, we also included a metric defined as follows:

$$\frac{\max \text{diff}(\mathbf{s})}{s_{max}}$$

where $\mathbf{s}$ is a vector of sorted ISIs, and $s_{max}$ is the sorted ISI for which the difference between it and the previous sorted ISI is maximum. This metric was included as it captures to a first approximation how 'burst-like' a spike train is. Intuitively, this metric is high for spike trains with bimodal ISI distributions, as is the case during bursts.

All these features were combined into a single data frame and $z$-scored (*Figure 2d*).

In some cases, these features were not defined, for example, when there are no spikes on either neuron, the concepts of spike phases or ISIs are meaningless. In these cases, 'filler' values were used that were located well off the extremes of the distribution of the metric when defined. For example, ISIs were filled with values of 20 s (the size of the bin) when no spikes were observed. The overall results and shape of the embedding did not depend sensitively on the value of the filler values used.

These features were chosen to capture various modes of spiking and bursting that have been previously identified by manual inspection (*Haddad and Marder, 2018*; *Tang et al., 2012*; *Haley et al., 2018*). Other features may be more appropriate in other systems where spike patterns span different axes of variability. However, we note that these features while being appropriate for this data were not 'fine-tuned' to specialize in features that are exclusively found in spike patterns from the pyloric circuit. For example, these features do not explicitly measure bursting, the dominant feature of the pyloric rhythm, but instead use distributions of ISIs that are sufficiently descriptive to capture the variability in bursting and transitions from bursting to other spiking.

## Embedding using t-SNE

So far, we have described how we converted a 20 s snippet containing spike times from LP and PD into a data frame (a vector). We did this for every 20 s snippet in the dataset. Data that did not fit into any bin was discarded (e.g., data at the trailing end of an experiment shorter than 20 s). Thus, our entire dataset is represented by $M \times N$ matrix, where $M$ is the number of features in the data frame and $N$ is the number of data points.

We used the t-SNE algorithm (*Van der Maaten and Hinton, 2008*) to visualize the vectorized data matrix in two dimensions. Our dataset contained $\approx 10^5$ points and was therefore too large for easy use of the original t-SNE algorithm. We used the FI-t-SNE approximate algorithm (*Linderman et al., 2019*) to generate these embeddings. We used a perplexity of $P = 100$ to generate these embeddings. Varying perplexity caused the embedding to change in ways consistent with what is expected for t-SNE embeddings, and the coarse features of the embedding did not sensitively depend on this choice of perplexity (*Figure 2—figure supplement 4*). t-SNE is often used with random initialization, and different random initializations can lead to different embeddings with clusters located at different positions in the map. The importance of meaningful initializations has recently been highlighted (*Kobak and Linderman, 2021*), and we used a fixed initialization where the x-axis corresponded to the shortest ISI in each data point and the y-axis corresponded to the maximum ratio of second-order to first-order ISI ratios (described above). For completeness, we also generated embeddings using other initializations (*Figure 3—figure supplement 2*). For both random initializations (*Figure 3—figure supplement 2a–d*) and initializations based on ISIs (*Figure 3—figure supplement 2e and f*), we observed that regular states tended to occur in a single region, surrounded by clusters that were dominated by a single color corresponding to irregular states. Thus, the precise location of different clusters can vary with the initialization, but the overall structure of the embedding, and the identity of points that tend to co-occur in a cluster, does not vary substantially with initialization.

## Manual clustering and annotation of data

Once the feature vectors were embedded using t-SNE, we manually inspected these points to get a sense of the spike patterns in each point cloud. To do so, we built an interactive tool that visualized spike patterns that corresponded to each point when clicked on. Random points within regions of high density were sampled to check that interior points had similar spike patterns. Points were assigned labels by drawing boundaries around them and labeling all points within that boundary. Finally, we generated plots of ISIs and rasters from points in clusters to ensure that patterns of spiking were visually similar.

### Triangulation and triadic differences (Figure 2—figure supplement 3)

The output of the embedding algorithm is a set of points in two dimensions. We built a Delaunay triangulation on these points. For each triangle in the triangulation, we computed the maximum difference between some burst metric (e.g., burst period of PD neurons) across the three vertices of that triangle. These triadic differences are represented colored dots, where the dots are located at the incenters of each triangle in the triangulation.

## Time-series analysis

### Measuring burst metrics (Figure 4)

Burst metrics were measured following previous definitions (*Prinz et al., 2004*; *Bucher et al., 2005*). Briefly, bursts were identified by observing that ISI distributions were bimodal, with smaller ISIs corresponding to ISIs within a burst, and longer ISIs corresponding to inter-burst intervals. This allowed us to threshold ISIs, and this identifies burst starts and burst ends. From here, burst periods could be calculated, which allowed us to measure phases and delays relative to the start of the PD burst.

### Measuring transition matrices (Figures 6, 7 and 10)

The transition matrix is a square matrix of size $N$ that describes the probability of transitioning from one to another of $N$ possible states. The transition matrix we report is the right stochastic matrix, where rows sum to 1. Each element of the matrix $T_{ij}$ corresponds to the conditional probability that we observe state $j$ given state . To compute this, we iterate over the sequence of states and compare the current state to the state in the next state. Breakpoints in the sequence are identified by discontinuities in the timestamps of that sequence and are ignored. We then zeroed the diagonal of the matrix and normalized each row by the sum.

### Measuring variability before transitions away from regular states (Figures 6 and 7)

We first identified continuous segments that corresponded to uninterrupted recordings from the same preparation at the appropriate condition. For each segment, we found all transitions away from the regular state. We therefore computed a vector as long as the segment containing the time to the next transition. We then collected points corresponding to time to next transition ranging from $t = -200s$ to $t = 0s$. For each time bin, we measured the CV of the burst period by dividing the standard deviation of the burst period in that datum by the mean in that datum.

## Data visualization

### Raincloud plots (Figure 4)

Raincloud plots (*Allen et al., 2019*) are used to visualize a univariate distribution. Individual points are plotted as dots, and a shaded region indicates the overall shape of the distribution. This shape is obtained by estimating a kernel smoothing function estimate over the data. Individual points are randomly jittered along the vertical axis for visibility.

### Occupancy maps (Figures 5 and 7)

To visualize where in the map data from a certain condition occurred, the full embedding is first plotted with colors corresponding to the state each point belongs to. The full dataset is made semi-transparent and plotted with larger dots to emphasize the data of interest. Data in the condition of interest is then plotted as usual. Each bright point in these plots corresponds to a 20 s snippet of data in the condition indicated.

### Treemaps (Figures 7 and 9)

Treemaps (*Shneiderman and Wattenberg, 2001*) were used to visualize state probabilities in a given experimental condition. For each preparation, the probability of each state was computed, and the mean probability of a given state was computed by averaging across all preparations. Thus, each preparation contributes equally. The area of the region in the treemap scales with the probability of that state.

## Transition matrices (Figures 6, 7 and 10)

Transition matrices were visualized as in *Corver et al., 2021*. Initial states are shown along the left edge, and final states are shown along the bottom edge of each matrix. Lines are colored by origin (horizontal lines) or destination (vertical) states. The size of each disc at the intersection of each line scales with the conditional probability of moving from the initial state to the final state. Note that the size of all discs is offset by a constant to make small discs visible.

## Statistics

### Comparing within-group to across-group variability (Figure 4)

To compare the variability of various burst metrics within each animal and across animals, we first measured the means and CVs of each burst metrics in every animal. We then used the mean of the coefficients of variations as a proxy for the within-animal variability and used the CV of the means as a proxy for the across-animal variability. Note that both measures are dimensionless. They can therefore be directly compared.

To test if the within-animal variability was significantly less than the across-animal variability, we performed a permutation test. We shuffled the labels identifying the animal to which each data point belonged to and measured a new 'within-animal' and 'across-animal' variability measure using these shuffled labels. We repeated this process 1000 times to obtain a null distribution of differences between within- and across-animal variability. Identifying where in the null distribution the data occurred allowed us to estimate a p-value for the measured difference. For example, if the measured difference between within- and across-animal variability in metric X was greater than 99% of the null distribution obtained by shuffling labels, we conclude that the p-value is 0.01. The significance level of 0.05 was divided by the number of burst metrics we tested to determine if any one metric was significantly more or less variable across animals.

### Measuring trends in variability in regular rhythms before transitions (Figures 6b, 7f and 9d)

To determine if variability significantly increased in the 200 s preceding a transition away from regular, we measured the Spearman rank correlation between time before transition (x-axis) and mean variability. The Spearman rank correlation $\rho$ is 1 if quantities monotonically increase.

### Measuring transition rate significance (Figures 6a, 7e and 10)

In the empirical transition matrices, certain transitions never occur, and certain transitions occur with relatively high probability. Each element of the transition matrix $T_{ij}$ corresponds to the conditional probability $P(\text{final}|\text{initial})$. Our null model assumes that transitions occur at random between states, and therefore the probability of observing any transition $i \to j$ scales with the marginal probability of observing state $j$ after transitions. We therefore built a null distribution of transition rates by sampling with replacement from the marginal counts of states after transitions. The fraction of this null distribution that was above or below the empirical transition rate was interpreted to be the p-value and thus determined significance.

## Code availability

*Table 4* lists the code used in this article. The code can be downloaded by prefixing https://github.com/ to the project name.

**Table 4.** Code availability.

| Project | Notes |
| --- | --- |
| sg-s/crabsort | Interactive toolbox to sort spikes from extracellular data |
| sg-s/stg-embedding | Contains all scripts used to generate every figure in this article |
| KlugerLab/FIt-SNE | Fast interpolation-based *t*-distributed stochastic neighbor embedding, used to make embedding |
| sg-s/SeaSurfaceTemperature | Wrapper to scrape NOAA databases |

## Acknowledgements

This article includes data collected by Lamont Tang, Lily He, Mara Rue, Jessica Haley, Daniel Powell, Anatoly Rinberg, and Ekaterina Morozova. We gratefully acknowledge helpful conversations with Paul Miller, Mark Zielinksi, Sriram Sampath, and Alec Hoyland. This work was funded by NIH grants T32 NS007292 (SGS) and R35 NS097343 (EM and SGS), NIH MH060605 (FN and DB), and DFG SCHN 1594/1-1 (ACS).

## Additional information

### Funding

| Funder | Grant reference number | Author |
|---|---|---|
| National Institutes of Health | T32 NS007292 | Srinivas Gorur-Shandilya |
| National Institutes of Health | R35 NS097343 | Srinivas Gorur-Shandilya Eve Marder |
| National Institutes of Health | MH060605 | Dirk M Bucher Farzan Nadim |
| Deutsche Forschungsgemeinschaft | DFG SCHN 1594/1-1 | Anna C Schneider |

The funders had no role in study design, data collection and interpretation, or the decision to submit the work for publication.

### Author contributions

Srinivas Gorur-Shandilya, Conceptualization, Data curation, Formal analysis, Investigation, Methodology, Software, Validation, Visualization, Writing – original draft, Writing – review and editing; Elizabeth M Cronin, Investigation, Methodology, Writing – review and editing; Anna C Schneider, Investigation, Methodology, Resources, Writing – review and editing; Sara Ann Haddad, Conceptualization, Investigation, Methodology, Writing – review and editing; Philipp Rosenbaum, Investigation, Methodology; Dirk Bucher, Project administration, Software, Supervision, Writing – original draft, Writing – review and editing; Farzan Nadim, Project administration, Supervision, Writing – original draft, Writing – review and editing; Eve Marder, Conceptualization, Funding acquisition, Project administration, Supervision, Writing – original draft, Writing – review and editing

### Author ORCIDs

Srinivas Gorur-Shandilya http://orcid.org/0000-0002-7429-457X
Elizabeth M Cronin http://orcid.org/0000-0002-4949-0042
Anna C Schneider http://orcid.org/0000-0002-1270-836X
Sara Ann Haddad http://orcid.org/0000-0003-0807-0823
Philipp Rosenbaum http://orcid.org/0000-0002-9976-366X
Dirk Bucher http://orcid.org/0000-0003-4144-2895
Farzan Nadim http://orcid.org/0000-0003-4144-9042
Eve Marder http://orcid.org/0000-0001-9632-5448

### Decision letter and Author response

Decision letter https://doi.org/10.7554/eLife.76579.sa1
Author response https://doi.org/10.7554/eLife.76579.sa2

## Additional files

### Supplementary files
• Transparent reporting form
• Supplementary file 1. Interactive visualization of pyloric dynamics.

## Data availability

All data needed to reproduce figures in this paper are available at https://zenodo.org/record/5090130.

The following dataset was generated:

| Author(s) | Year | Dataset title | Dataset URL | Database and Identifier |
|-----------|------|---------------|-------------|------------------------|
| Srinivas Gorur-Shandilya Brandeis University ; Elizabeth M Cronin; Anna C. Schneider; Sara Ann Haddad; Philipp Rosenbaum; Dirk Bucher; Farzan Nadim; Eve Marder | 2021 | Mapping circuit dynamics during function and dysfunction | https://doi.org/10.5281/zenodo.5090130 | Zenodo, 10.5281/zenodo.5090130 |

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
