## [Editor Report]

This study applies an unsupervised dimensionality reduction (t-SNE) to characterize neural spiking dynamics in the pyloric circuit in the stomatogastric ganglion of the crab, an important system for mechanistic analysis of rhythmic circuit function. The application of unsupervised methods to characterize qualitatively distinct regimes of spiking neural circuits is interesting and novel. The challenges and lessons learned in this study are of broader interest to those seeking to quantitatively characterize large sets of neural data across many subjects. The method is demonstrated across hundreds of animal subjects and used to investigate circuit responses to a variety of perturbations.

---

## [Decision Letter]

**Decision letter after peer review:**

[Editors’ note: the authors submitted for reconsideration following the decision after peer review. What follows is the decision letter after the first round of review.]

Thank you for submitting the paper "Mapping Circuit Dynamics During Function and Dysfunction" for consideration by *eLife*. Your article has been reviewed by 3 peer reviewers, and the evaluation has been overseen by a Reviewing Editor and a Senior Editor. The following individuals involved in review of your submission have agreed to reveal their identity: Alex Williams (Reviewer #1).

We are sorry to say that, after consultation with the reviewers, we have decided that this work will not be considered further for publication by *eLife*.

The reviewers find that paper has intrinsic interest especially to specialists in motor pattern generation by small circuits but requires considerable work and refocusing to make it of broader significance.

The expert reviews are quite compatible and provide rich feedback that could lead to a major overhaul and a new submission to *eLife*.

1. t-SNR is not in itself a clustering algorithm so its resulting 'clusters' should be validated by application of a true clustering algorithm. t–SNE does often provide a nice 2–D embedding that can simplify subsequent clustering. Here the reviewers suggest that the feature vectors themselves should be subjected to clustering algorithm.

2. The feature vectors are not as well motivated and explained as they should be so that readers in other systems can use the strategy presented here in cases where other features than ISI and spike phase of the activity might be more relevant.

3. There was discussion as to whether the framing of the manuscript as a ML pipeline is appropriate because the pipeline described relies heavily on manual curation. The manual curation should be more explicitly described (with details), and the limitations of this approach – ML followed by manual curation – discussed. Such a pipeline probably can't be as easily downloaded and applied directly to someone else's data, but overall, the community could benefit from more examples/consideration of how to systematically and carefully interleave the use of machine–learning techniques with manual analysis by domain experts but the process should be more explicitly explained.

4. Discussion should be revamped to discuss more fully specific insights into the pyloric circuit and the limitations of the analyses presented, for example the omission of the PY neuron activity means that the map as given is incomplete: potentially there are many more states, and hence transitions, within or beyond those already found that correspond to changes in PY neuron activity.

*Reviewer #1:*

The authors sought to establish a standardized quantitative approach to categorize the activity patterns in a central pattern generator (specifically, the well–studied pyloric circuit in C. borealis). While it is easy to describe these patterns under "normal" conditions, this circuit displays a wide range of irregular behaviors under experimental perturbations. Characterizing and cataloguing these irregular behaviors is of interest to understand how the network avoids these dysfunctional patterns under "normal" circumstances.

The authors draw upon established machine learning tools to approach this problem. To do so, they must define a set of features that describe circuit activity at a moment in time. They use the distribution of inter–spike–intervals ISIs and spike phases of the LP and PD neuron as these features. As the authors mention in their Discussion section, these features are highly specialized and adapted to this particular circuit. This limits the applicability of their approach to other circuits with neurons that are unidentifiable or very large in number (the number of spike phase statistics grows quadratically with the number of neurons).

The main results of the paper provide evidence that ISIs and spike phase statistics provide a reasonable descriptive starting point for understanding the diversity of pyloric circuit patterns. The authors rely heavily on t–distributed stochastic neighbor embedding (tSNE), a well–known nonlinear dimensionality reduction method, to visualize activity patterns in a low–dimensional, 2D space. While effective, the outputs of tSNE have to be interpreted with great care (Wattenberg, et al., "How to Use t–SNE Effectively", Distill, 2016. http://doi.org/10.23915/distill.00002). I think the conclusions of this paper would be strengthened if additional machine learning models were applied to the ISI and spike phase features, and if those additional models validated the qualitative results shown by tSNE. For example, tSNE itself is not a clustering method, so applying clustering methods directly to the high–dimensional data features would be a useful validation of the apparent low–dimensional clusters shown in the figures.

The authors do show that the algorithmically defined clusters agree with expert–defined clusters. (Or, at least, they show that one can come up with reasonable post–hoc explanations and interpretations of each cluster). The very large cluster of "regular" patterns – shown typically in a shade of blue – actually looks like an archipelago of smaller clusters that the authors have reasoned should be lumped together. Thus, while the approach is still a useful data–driven tool, a non–trivial amount of expert knowledge is baked into the results. A central challenge in this line of research is to understand how sensitive the outcomes are to these modeling choices, and there is unlikely to be a definitive answer.

Nonetheless, the authors show results which suggest that this analysis framework may be useful for the community of researchers studying central pattern generators. They use their method to qualitatively characterize a variety of network perturbations – temperature changes, pH changes, decentralization, etc.

In some cases it is difficult to understand the level of certainty in these qualitative observations. A first look at Figure 5a suggests that three different kinds of perturbations push the circuit activity into different dysfunctional cluster regions. However, the apparent spatial differences between these three groups of perturbations might be due to animal–level differences (i.e. each preparation produces multiple points in the low–D plot, so the number of effective statistical replicates is smaller than it appears at first glance). Similarly, in Figure 9, it is somewhat hard to understand how much the state occupancy plots would change if more animals were collected –– with the exception of proctolin, there are ~25 animals and 12 circuit activity clusters which may not be a favorable ratio. It would be useful if a principled method for computing "error bars" on these occupancy diagrams could be developed. Similar "error bars" on the state transition diagrams (e.g. Figure 6a) would also be useful.

Finally, one nagging concern that I have is that the ISIs and spike phase statistics aren't the ideal features one would use to classify pyloric circuit behaviors. Sub–threshold dynamics are incredibly important for this circuit (e.g. due to electrical coupling of many neurons). A deeper discussion about what is potentially lost by only having access to the spikes would be useful.

Overall, I think this work provides a useful starting point for large–scale quantitative analysis of CPG circuit behaviors, but there are many additional hurdles to be overcome.

*Reviewer #2:*

This manuscript uses the t–SNE dimensionality reduction technique to capture the rich dynamics of the pyloric circuit of the crab.

Strengths:

– The integration of a rich data–set of spiking data from the pyloric circuit.

– Use of nonlinear dimension reduction (t–SNE) to visualise that data.

– Use of clusters from that t–SNE visualisation to create subsets of data that are amenable to consistent analyses (such as using the "regular" cluster as a basis for surveying the types of dynamics possible in baseline conditions).

– Innovative use of the cluster types to describe transitions between dynamics within the baseline state and within perturbed states (whether by changes to exogenous variables, cutting nerves, or applying neuromodulators).

Some interesting main results:

– Baseline variability in the spiking patterns of the pyloric circuit is greater within than between animals.

– Transitions to silent states often (always?) pass through the same intermediate state of the LP neuron skipping spikes.

Weaknesses:

– t-SNE is not, in isolation, a clustering algorithm, yet here it is treated as such. How the clusters were identified is unclear: the manuscript mentions manual curation of randomly sampled points, implying that the clusters were extrapolations from these. This would seem to rather defeat the point of using unsupervised techniques to obtain an unbiased survey of the spiking dynamics and raises the issue of how robust the clusters are.

– The main purpose and contribution of the paper is unclear, as the results are descriptive, and mostly state that dynamics in some vary between different states of the circuit; while the collated dataset is a wonderful resource, and the map is no doubt useful for the lab to place in context what they are looking at, it is not clear what we learn about the pyloric circuit, or more widely about the dynamical repertoire of neural circuits.

– In some places the contribution is noted as being the pipeline of analysis: unfortunately as the pipeline used here seems to rely in manual curation, it is of limited general use; moreover, there are already a number of previous works that use unsupervised machine–learning pipelines to characterise the complexity of spiking activity across a large data–set of neurons, using the same general approach here (quantify properties of spiking as a vector; map/cluster using dimension reduction), including Baden et al. (2016, Nature), Bruno et al. (2015, Neuron), Frady et al. (2016, Neural Computation).

Some key limitations are not considered:

– The omission of the PY neuron activity means that the map as given is incomplete: potentially there are many more states, and hence transitions, within or beyond those already found that correspond to changes in PY neuron activity.

– The use of long, non–overlapping time segments (20s) – this means, for example, that the transitions are slow and discrete, whereas in reality they may be abrupt, or continuous.

– tSNE cannot capture hierarchical structure, nor has a null model to demonstrate that the underlying data contains some clustering structure. So, for example, distances measured on the map may not be strictly meaningful if the data is hierarchical.

– The Discussion does not include enough insight and contextualisation of the results.

Recommendations:

Explain and validate the clusters:

– The paper explains only that the clusters were assigned by manual curation of randomly sampled points. How then were all points assigned to clusters given that sample? What metric was used?

– We would suggest validating the clusters by using a clustering algorithm to recover them (approximately) from the vectors

Clarify the paper's goals and contributions:

– As noted above, the use of an unsupervised pipeline to map spiking activity is not novel; the survey of data is interesting, but purely descriptive. The key contributions were then not obvious to this reader – please clarify. To give some suggestions, while reading 3 possible contributions came to mind, but none quite fit: if the paper is an announcement of the dataset, then the datasets need a detailed explanation; if the paper is the introduction of a pipeline, then the pipeline ought to be fully unsupervised, else it is of little use outside the hands of the present lab; if the paper is about insights into the pyloric circuit, then conclusions and insights ought to be drawn from the descriptive results

Improve the Discussion:

– Explicitly link the results to insights into the pyloric circuit. The abstract states "strong mechanistically interpretable links between complex changes in the collective behavior of a neural circuit and specific experimental manipulations" – but only the short section on "Diversity and stereotypy in trajectories from functional to crash states" touches on this.

– Long section on other clustering methods is a survey of some alternatives and draws no conclusions about why tSNE was chosen here: either use to justify tSNE, or omit.

– Lines 678–716 are not based on any results in the paper.

Discuss limitations:

– The omission of the PY neuron activity means that the map as given is incomplete – what might it look like with it (new states, new transitions etc)?

– The transitions are slow and discrete by design (20 s long segments), whereas in reality they may be abrupt, or continuous.

– tSNE's limitations and their implications e.g. it cannot capture hierarchical structure, nor has a null model to demonstrate that the underlying data contains some clustering structure.

– Why those features of the spike–trains, and how would the map change if features were omitted or added? (e.g. the regularity of spiking).

*Reviewer #3:*

Gorur–Shandilya et al. apply an unsupervised dimensionality reduction (t–SNE) to characterize neural spiking dynamics in the pyloric circuit in the stomatogastric ganglion of the crab. The application of unsupervised methods to characterize qualitatively distinct regimes of spiking neural circuits is very interesting and novel, and the manuscript provides a comprehensive demonstration of its utility by analyzing dynamical variability in function and dysfunction in an important rhythm–generating circuit. The system is highly tractable with small numbers of neurons, and the study here provides an important new characterization of the system that can be used to further understand the mapping between gene expression, circuit activity, and functional regimes. The explicit note about the importance of visualization and manual labeling was also nice, since this is often brushed under the rug in other studies.

While the specific analysis pipeline clearly identifies qualitatively distinct regimes of spike patterns in the LP/PD neurons, it is not clear how much of this is due to t–SNE itself vs the initial pre–processing and feature definition (ISI and spike phase percentiles). Analyses that would help clarify this would be to check whether the same clusters emerge after (1) applying ordinary PCA to the feature vectors and plotting the projections of the data along the first two PCs, or (2) defining input features as the concatenated binned spike rates over time of the LP & PD neurons (which would also yield a fixed–length vector per 20 s trial), and then passing these inputs to PCA or t–SNE. As the significance of this work is largely motivated by using unsupervised vs ad hoc descriptors of circuit dynamics, it will be important to clarify how much of the results derive from the use of ISI and phase representation percentiles, etc. as input features, vs how much emerge from the dimensionality reduction.

Please define all acronyms when they are introduced.

Why is 20 seconds chosen as the time window to compute ISIs/phase representations before feeding the data into t–SNE? Is this roughly the window one needs to look at to visually identify differences across dynamical regimes and can the authors explain why?

It would be useful to note in the manuscript that t–SNE is tuned to preserve local over global similarities, which gives it its utility in clustering. It would also be useful to discuss the relationship of t–SNE to UMAP (McInnes, Healy, Melville 2018 arXiv: https://arxiv.org/abs/1802.03426), another popular dimensionality reduction technique in neuroscience.

Most uses of t–SNE in biology in the past have been in the context of classifying behavior (e.g. Berman et al. 2014 J R Soc Interface: https://www.ncbi.nlm.nih.gov/pmc/articles/PMC4233753/ or Clemens et al. Current Biology 2018: https://www.sciencedirect.com/science/article/pii/S0960982218307735). Including these references would be helpful as a point of comparison and to emphasize the novelty of applying t–SNE to neural spiking data.

Can the authors comment on how much pyloric rhythms can deviate from the standard triphasic pattern before behavior (or digestion?) is significantly disrupted?

In Figure 6 it would be helpful to show example time–series going into the transition matrix analysis, with identified state labels at the different timepoints. It wasn't clear whether timepoints were e.g. moment–to–moment, 20 sec chunks, a 20 sec sliding window, etc.

In Figure 8 it would be helpful to show examples of what bursts look like with and without decentralization, in order to contextualize the quantitative metrics.

It would also be useful to connect this work in the discussion to modern approaches for identifying model parameter regimes underlying distinct neural activity patterns (see e.g. Bittner et al. 2021 bioRxiv https://www.biorxiv.org/content/10.1101/837567v3)

[Editors’ note: further revisions were suggested prior to acceptance, as described below.]

Thank you for resubmitting your work entitled "Mapping circuit dynamics during function and dysfunction" for further consideration by *eLife*. Your revised article has been evaluated by Ronald Calabrese (Senior Editor) and a Reviewing Editor.

The manuscript has been improved but there are some remaining issues that need to be addressed, as outlined below:

Two reviewer concerns remain, which are explained more fully in the reviews below.

1. The analysis pipeline outlined in this paper is rather ad hoc. This isn't a general "tool" as it has only been demonstrated in one particular biological system.

2. It is acceptable for the authors to call the core method (tSNE) unsupervised. The "manual curation" is part of any exploratory data analysis, though it is generally preferable to for this manual curation / interpretation step to be as objective and reproducible as possible (which is arguably not entirely achieved in this paper). The paper can be seen as an investigation into how to apply an existing tool (tSNE) to a very well-studied and simple neural system.

In revision, we ask the authors to take a final pass through the Introduction and Discussion and to tone down any comments about this being a general, off-the-shelf tool and to note that while the core algorithmic method (tSNE) is unsupervised, there is still the need for expert interpretation of the output.

*Reviewer #1:*

The manuscript has been revised along the lines suggested in both the consensus suggestions and the individual reports. The rewritten Discussion is considerably improved. There is a lot of hard work here, and I've no doubt the results are useful to the PI's lab, and presumably other workers on this CPG.

To my mind the authors did not really address two of the main concerns of the reviewers: (1) what the purpose of this paper is and (2) the validity of the manual clustering of tSNE. In more detail:

Purpose of the paper: the paper prominently refers to it introducing an "unsupervised" method/approach (e.g. lines 69-70) and a "tool" (lines 537-541, 711-714). Neither are true: the approach here is by definition not unsupervised as it uses manual curation to identify the clusters upon which all further analyses are based; such manual curation means that the only "tool" parts are using tSNE on a feature vector, which doesn't merit the term. Rather, it seems that the purpose of the paper is instead to provide a cohesive overview of the dynamics of this specific CPG system, by finding a way to systematically combine a large number of separate recordings from baseline and perturbed preparations, thus allowing knowledge discovery (e.g. the types of transitions; the changes in the regular state that precede transitions) on those combined data.

tSNE and validating the clustering: the authors did considerable work related to this issue, by using their approach on synthetic data, replacing tSNE with UMAP, and by testing simple hierarchical and k-means clustering of their feature vectors. But none of these spoke directly to the validity of the clustering: the synthetic data and UMAP were still manually clustered, so could be made to conform to the desired results; the unsupervised clustering approach used shows the well-known problem that directly clustering a high-dimensional dataset (a 48-dimensional feature vector) is often impossible using classical techniques simply due to the curse of dimensionality. The normal approach here would be to use dimension reduction then unsupervised clustering (e.g. in the classic combination of PCA then k-means), matching the way the two steps are done in the present paper.

As the paper shows, the clusters found in tSNE by manual curation are arbitrary. For example, changing the perplexity parameter changes the fragmentation of the tSNE map, so would likely lead to different manual clusterings. In another example, the clusters found by manual curation are not all contiguous in the 2D space e.g. the LP-silent cluster is in 3 groups, one of which is on the opposite side of the 2D space to the others – so distance in this space is not interpreted as meaningful for all clusters. In another example, the dividing line between some "clusters" is arbitrarily put in a contiguous run of points – e.g. the LP-weak-skipped (pink) and irregular-bursting (brown).

So to me it seems we're still left with the original main issues raised, and consequently am still unclear who this paper is aimed at – this is a useful way for the lab who owns that data to organise it, but how useful is it in revealing the structure of dynamics in the CPG, and hence to others? Moreover, though the Discussion is much improved, the paper is still rather descriptive throughout.

These are clearly subjective concerns, not issues with the quality of the analyses done based on the clustering, which are solid, nor with their interpretation, which has been considerably improved. Nonetheless, I think the paper would still be considerably improved by a concerted redraft, and perhaps reorganisation, to deal with these issues.

*Reviewer #2:*

I appreciate that the authors carefully considered my review and tried new analyses (e.g. hierarchical clustering and UMAP) and have expanded the Discussion section to comment on the issues raised during review. I also agree that "a rigorous error analysis would be useful but is not trivially done" -- and since I do not have a concrete suggestion here, I think the manuscript should be published without one. Overall, I feel that my concerns have been satisfactorily addressed.

*Reviewer #3:*

One small remaining edit that would be helpful is to show the projection of the data onto the first two PCs (re: the PCA analysis referenced in Figure 2-Figure supp 2), so that the reader can visualize what structure there is prior to computing the triadic difference statistics. If not too computationally intensive, it would also be useful to show what emerges from applying a naive clustering or visualization algorithm directly on the spike-train time-series [see (2) in Reviewer #3 major concern], even if only to show that it is quite messy.

---

## [Author Response]

[Editors’ note: the authors resubmitted a revised version of the paper for consideration. What follows is the authors’ response to the first round of review.]

1. t-SNR is not in itself a clustering algorithm so its resulting 'clusters' should be validated by application of a true clustering algorithm. t–SNE does often provide a nice 2–D embedding that can simplify subsequent clustering. Here the reviewers suggest that the feature vectors themselves should be subjected to clustering algorithm.

We agree with the reviewers is t-SNE is not in itself a clustering algorithm. We have been careful to use it as a *visualization* and *data reduction* tool, whose strengths are uniquely suited to the task we applied it to. The only clustering we performed in the original manuscript was entirely manual, using the t-SNE embedding to visualize the data and a tool to quickly inspect the raw spike patterns to guide our clustering. As the reviewers point out, “t-SNE does often provide a nice 2-D embedding that can simplify subsequent clustering”. This is exactly what we did. The strength of our approach is that it made feasible a previously intractable task: manually classifying hundreds of hours of spike patterns from hundreds of animals.

We agree with the reviewers that this paper can be strengthened by validation of this technique and by also comparing to alternative techniques. We therefore did the following:

1. We created a synthetic data set with spikes from two neurons drawn from different patterns. We then subjected this data set to the workflow described in the manuscript and found we could recover distinct classes of spike patterns that existed in the synthetic data. This new analysis is shown in Figure 2—figure supplement 4.

2. In addition, we used another nonlinear dimensionality reduction technique (uMAP) to also embed the feature vectors in two-dimensional space. We do this to emphasize that t-SNE is only a tool to visualize patterns in high dimensional data and is not the only one that can be useful, and that clustering of this data comes not from t-SNE but is a distinct step following embedding (Figure 3— figure supplement 3).

3. Finally, we use hierarchical clustering to directly cluster the feature vectors. In Author response image 1, (a) shows how the number of clusters varies with the cutoff chosen. Allowing the cutoff to vary to yield a maximum of 20 clusters, we color the points in the t-SNE embedding by cluster identity (b). The vast majority of the data are grouped into a single cluster, with disparate spike patterns grouped together. This is likely because we used Euclidean distances to determine linkage between clusters, and clustering may be dominated by global rather than local structure in the feature vectors. A similar picture emerges when we use *k*means clustering on the feature vectors (c) Setting the number of clusters *k* = 20. Cluster boundaries do not correspond to sharp discontinuities in spike patterns, as observed by manual inspection. (d) shows the distribution of points in each cluster. These results demonstrate that the methods we propose offer utility beyond naïve hierarchical or *k*-means clustering and are better suited to recover patterns in neural data.

**Author response image 1. sa2fig1:** 

2. The feature vectors are not as well motivated and explained as they should be so that readers in other systems can use the strategy presented here in cases where other features than ISI and spike phase of the activity might be more relevant.

We have expanded the description of the feature vectors in the Methods:

Construction of vectorized data frame to go into greater detail. We have included a discussion of why we chose these feature vectors, and proposed alternatives to ease applicability in other systems.

3. There was discussion as to whether the framing of the manuscript as a ML pipeline is appropriate, because the pipeline described relies heavily on manual curation. The manual curation should be more explicitly described (with details), and the limitations of this approach – ML followed by manual curation – discussed. Such a pipeline probably can't be as easily downloaded and applied directly to someone else's data, but overall, the community could benefit from more examples/consideration of how to systematically and carefully interleave the use of machine–learning techniques with manual analysis by domain experts but the process should be more explicitly explained.

The method described here uses a nonlinear dimensionality reduction algorithm to generate a two-dimensional visualization that allows a domain expert to efficiently label and cluster large amounts of data. We agree that this is a manual process that requires human intervention. We have extensively rewritten the Discussion to take this into account and included a section “Applicability to other systems” that goes into detail what needs to be done to apply this method to other circuits. We have included a statement in the Discussion making clear that a drawback of this method is that it is not fully automated and requires a human to inspect data and draw cluster boundaries (lines 532-541).

4. Discussion should be revamped to discuss more fully specific insights into the pyloric circuit and the limitations of the analyses presented, for example the omission of the PY neuron activity means that the map as given is incomplete: potentially there are many more states, and hence transitions, within or beyond those already found that correspond to changes in PY neuron activity.

We agree with the reviewers that the omission of the PY neurons’ activity means that the map is incomplete. There are likely many more states, and hence many more transitions, than the ones we have identified. In addition, we note that there are other pyloric neurons whose activity is also missing (AB, IC, LPG, VD). However, measuring just LP and PD allows us to monitor the activity of the most important functional antagonists in the system (because they are effectively in a half-center oscillator because PD is electrically coupled to AB). In general, the more neurons one measures, the richer the description of the circuit dynamics will be, and the more transitions one will observe. Collecting datasets at this scale (~500 animals) from all pyloric neurons is challenging, and we have revised the manuscript to make this important point (see Discussion: Technical considerations).

Reviewer #1:The authors sought to establish a standardized quantitative approach to categorize the activity patterns in a central pattern generator (specifically, the well–studied pyloric circuit in C. borealis). While it is easy to describe these patterns under "normal" conditions, this circuit displays a wide range of irregular behaviors under experimental perturbations. Characterizing and cataloguing these irregular behaviors is of interest to understand how the network avoids these dysfunctional patterns under "normal" circumstances.The authors draw upon established machine learning tools to approach this problem. To do so, they must define a set of features that describe circuit activity at a moment in time. They use the distribution of inter–spike–intervals ISIs and spike phases of the LP and PD neuron as these features. As the authors mention in their Discussion section, these features are highly specialized and adapted to this particular circuit. This limits the applicability of their approach to other circuits with neurons that are unidentifiable or very large in number (the number of spike phase statistics grows quadratically with the number of neurons).

We agree with the reviewer that the size of the feature vectors as described grows quadratically with the number of neurons. The feature sets we describe are most suited for “identified” neurons – neurons whose identity and connectivity are known and can be reliably recorded from multiple animals. The method described here is best suited for systems with small numbers of identified neurons. For other systems, other feature vectors may be chosen, as we have suggested in the Discussion: Applicability to other systems.

The main results of the paper provide evidence that ISIs and spike phase statistics provide a reasonable descriptive starting point for understanding the diversity of pyloric circuit patterns. The authors rely heavily on t–distributed stochastic neighbor embedding (tSNE), a well–known nonlinear dimensionality reduction method, to visualize activity patterns in a low–dimensional, 2D space. While effective, the outputs of tSNE have to be interpreted with great care (Wattenberg, et al., "How to Use t–SNE Effectively", Distill, 2016. http://doi.org/10.23915/distill.00002). I think the conclusions of this paper would be strengthened if additional machine learning models were applied to the ISI and spike phase features, and if those additional models validated the qualitative results shown by tSNE. For example, tSNE itself is not a clustering method, so applying clustering methods directly to the high–dimensional data features would be a useful validation of the apparent low–dimensional clusters shown in the figures.

We thank the reviewer for these suggestions and agree with the reviewer that t-SNE is not a clustering method, and directly clustering on t-SNE embeddings is rife with complexities. Instead, we have used t-SNE to generate a visualization that allows domain experts to quickly label and cluster large quantities of data. This makes a previously intractable task feasible, and offers some basic guarantees on quality (e.g., no one data point can have two labels, because labels derive from position of data points in two-dimensional space). In addition:

– We used uMAP, another dimensionality reduction algorithm, to perform the embedding step, and colored points by the original t-SNE embedding. (Figure 3—figure supplement 3). Large sections of the map are still strikingly colored in single colors, suggesting that the manual clustering did not depend on the details of the t-SNE algorithm, but is rather informed by the statistics of the data.

We validated our method using synthetic data. We generated synthetic spike trains from different “classes” and embedded the resultant feature vectors using t-SNE. Data from different classes are not intermingled and form tight “clusters” (Figure 2 —figure supplement 4).

– Finally, we attempted to use hierarchical clustering to cluster the raw feature vectors and were not able to find a reasonable portioning of the linkage tree that separated qualitatively different spike patterns (Figure at the top of this document). We speculate that this is because feature vectors may contain outliers that bias clustering algorithms that attempt to preserve global distance to lump the majority of the data into a single cluster, in order to differentiate outliers from the bulk of the data.

The authors do show that the algorithmically defined clusters agree with expert–defined clusters. (Or, at least, they show that one can come up with reasonable post–hoc explanations and interpretations of each cluster). The very large cluster of "regular" patterns – shown typically in a shade of blue – actually looks like an archipelago of smaller clusters that the authors have reasoned should be lumped together. Thus, while the approach is still a useful data–driven tool, a non–trivial amount of expert knowledge is baked into the results. A central challenge in this line of research is to understand how sensitive the outcomes are to these modeling choices, and there is unlikely to be a definitive answer.

We agree with the reviewer entirely.

Nonetheless, the authors show results which suggest that this analysis framework may be useful for the community of researchers studying central pattern generators. They use their method to qualitatively characterize a variety of network perturbations – temperature changes, pH changes, decentralization, etc.In some cases it is difficult to understand the level of certainty in these qualitative observations. A first look at Figure 5a suggests that three different kinds of perturbations push the circuit activity into different dysfunctional cluster regions. However, the apparent spatial differences between these three groups of perturbations might be due to animal–level differences (i.e. each preparation produces multiple points in the low–D plot, so the number of effective statistical replicates is smaller than it appears at first glance). Similarly, in Figure 9, it is somewhat hard to understand how much the state occupancy plots would change if more animals were collected –– with the exception of proctolin, there are ~25 animals and 12 circuit activity clusters which may not be a favorable ratio. It would be useful if a principled method for computing "error bars" on these occupancy diagrams could be developed. Similar "error bars" on the state transition diagrams (e.g. Figure 6a) would also be useful.

We agree with the reviewer. Despite this paper containing data from hundreds of animals, the dataset may not be sufficiently large to perform some necessary statistical checks. We agree with the reviewer that a more rigorous error analysis would be useful but is not trivially done.

Finally, one nagging concern that I have is that the ISIs and spike phase statistics aren't the ideal features one would use to classify pyloric circuit behaviors. Sub–threshold dynamics are incredibly important for this circuit (e.g. due to electrical coupling of many neurons). A deeper discussion about what is potentially lost by only having access to the spikes would be useful.

We agree with the reviewer that spike times aren’t the ideal feature to use to describe circuit dynamics. This is especially true in the STG, where synapses are graded, and coupling between cells can persist without spiking. However, the data required simply do not exist, as it requires intracellular recordings, which are substantially harder to perform (and maintain over challenging perturbations) than extracellular recordings.

Finally, the signal to the muscles – arguably the physiologically and functionally relevant signal – is the spike signal, suggesting that spike patterns from the pyloric circuit are a useful feature to measure. Nevertheless, this is an important point, and we thank the reviewer for raising it, and we have included it in the section titled Discussion: Technical considerations.

Overall, I think this work provides a useful starting point for large–scale quantitative analysis of CPG circuit behaviors, but there are many additional hurdles to be overcome.Reviewer #2:This manuscript uses the t–SNE dimensionality reduction technique to capture the rich dynamics of the pyloric circuit of the crab.Strengths:– The integration of a rich data–set of spiking data from the pyloric circuit.– Use of nonlinear dimension reduction (t–SNE) to visualise that data.– Use of clusters from that t–SNE visualisation to create subsets of data that are amenable to consistent analyses (such as using the "regular" cluster as a basis for surveying the types of dynamics possible in baseline conditions).– Innovative use of the cluster types to describe transitions between dynamics within the baseline state and within perturbed states (whether by changes to exogenous variables, cutting nerves, or applying neuromodulators).Some interesting main results:– Baseline variability in the spiking patterns of the pyloric circuit is greater within than between animals.– Transitions to silent states often (always?) pass through the same intermediate state of the LP neuron skipping spikesWeaknesses:– t-SNE is not, in isolation, a clustering algorithm, yet here it is treated as such. How the clusters were identified is unclear: the manuscript mentions manual curation of randomly sampled points, implying that the clusters were extrapolations from these. This would seem to rather defeat the point of using unsupervised techniques to obtain an unbiased survey of the spiking dynamics and raises the issue of how robust the clusters are.

We have used t-SNE to visualize the circuit dynamics in a two-dimensional map. We have exploited t-SNE’s ability to preserve local structure to generate an embedding where a domain expert can efficiently manually identify and label stereotyped clusters of activity. As the author points out, this is a manual step, and we have emphasized this in the manuscript. The strength of our approach is to combine the power of a nonlinear dimensionality reduction technique such as t-SNE with human curation to make a task that was previously impossible (identifying and labelling very large datasets of neural activity) feasible.

To address the question of how robust the manually identified clusters are, we have:

1. Used another dimensionality reduction technique, uMAP, to generate an embedding and colored points by the original t-SNE map (Figure 3 —figure supplement 3). To rough approximation, the coloring reveals that a similar clustering exists in this uMAP embedding.

2. We generated synthetic spike trains from pre-determined spike pattern classes and used the feature vector extraction and t-SNE embedding procedure as described in the paper. We found that this generated a map (Figure 2—figure supplement 4) where classes of spike patterns were well separated in the t-SNE space.

– The main purpose and contribution of the paper is unclear, as the results are descriptive, and mostly state that dynamics in some vary between different states of the circuit; while the collated dataset is a wonderful resource, and the map is no doubt useful for the lab to place in context what they are looking at, it is not clear what we learn about the pyloric circuit, or more widely about the dynamical repertoire of neural circuits.– In some places the contribution is noted as being the pipeline of analysis: unfortunately as the pipeline used here seems to rely in manual curation, it is of limited general use; moreover, there are already a number of previous works that use unsupervised machine–learning pipelines to characterise the complexity of spiking activity across a large data–set of neurons, using the same general approach here (quantify properties of spiking as a vector; map/cluster using dimension reduction), including Baden et al. (2016, Nature), Bruno et al. (2015, Neuron), Frady et al. (2016, Neural Computation).Some key limitations are not considered:– The omission of the PY neuron activity means that the map as given is incomplete: potentially there are many more states, and hence transitions, within or beyond those already found that correspond to changes in PY neuron activity.

We agree with the reviewer that the omission of the PY neurons’ activity means that the map is incomplete. There are likely many more states, and hence many more transitions, than the ones we have identified. In addition, we note that there are other pyloric neurons whose activity is also missing (AB, IC, LPG, VD). However, measuring just LP and PD allows us to monitor the activity of the most important functional antagonists in the system (because they are effectively in a half-center oscillator because PD is electrically coupled to AB). In general, the more neurons one measures, the richer the description of the circuit dynamics will be. Collecting datasets at this scale (~500 animals) from all pyloric neurons is challenging, and we have revised the manuscript to make this important point (see Discussion: Technical considerations).

– The use of long, non–overlapping time segments (20s) – this means, for example, that the transitions are slow and discrete, whereas in reality they may be abrupt, or continuous.

We agree with the reviewer. There are tradeoffs in choosing a bin size in analyzing time series – choosing longer bins can increase the number of “states” and choosing shorter bins can increase the number of transitions. We chose 20s bins because it is long enough to include several cycles of the pyloric rhythm, even when decentralized, yet was short enough to resolve slow changes in spiking. We have included a statement clarifying this (see Discussion: Technical considerations).

– tSNE cannot capture hierarchical structure, nor has a null model to demonstrate that the underlying data contains some clustering structure. So, for example, distances measured on the map may not be strictly meaningful if the data is hierarchical.

We agree with the reviewer. t-SNE can manifest clusters when none exist (Section 4 of https://distill.pub/2016/misread-tsne/) and can obscure or merge true clusters. We have restricted analyses that rely on distances measured in the map to cases where there are qualitative differences in behavior (e.g., with decentralization, Figure 7) or have compared distances within subsets of data where a single parameter is changed (e.g., pH or temperature, Figure 5). The only conclusion we draw from these distance measures is that data are more (or less) spread out in the map, which we use as a proxy for variability. We have included a statement, discussion limitations of using t-SNE (Discussion: Comparison with other methods).

– The Discussion does not include enough insight and contextualisation of the results.

We have completely rewritten the discussion to address this.

Recommendations:Explain and validate the clusters:– The paper explains only that the clusters were assigned by manual curation of randomly sampled points. How then were all points assigned to clusters given that sample? What metric was used?

We have added a new section in the Methods (Manual clustering and annotation of data) that describes this process.

– We would suggest validating the clusters by using a clustering algorithm to recover them (approximately) from the vectors.

We have validated the clusters and the embedding algorithm by:

1. Creating a synthetic data set with spikes from two neurons drawn from different patterns. We then subjected this data set to the workflow described in the manuscript and found we could recover distinct classes of spike patterns that existed in the synthetic data. This new analysis is shown in Figure 2—figure supplement 4.

2. In addition, we used another nonlinear dimensionality reduction technique (uMAP) to also embed the feature vectors in two-dimensional space. We do this to emphasize that t-SNE is only a tool to visualize patterns in high dimensional data and is not the only one that can be useful, and that clustering of this data comes not from t-SNE but is a distinct step following embedding. (Figure 3— figure supplement 3)

3. Finally, we use hierarchical clustering to directly cluster the feature vectors. Author response image 1, (a) shows how the number of clusters varies with the cutoff chosen. Allowing the cutoff to vary to yield a maximum of 20 clusters, we color the points in the t-SNE embedding by cluster identity (b). The vast majority of the data is grouped into a single cluster, with disparate spike patterns grouped together. This is likely because we used Euclidean distances to determine linkage between clusters, and clustering may be dominated by global rather than local structure in the feature vectors. A similar picture emerges when we use kmeans clustering on the feature vectors (c) setting the number of clusters k = 20. Cluster boundaries do not correspond to sharp discontinuities in spike patterns, as observed by manual inspection. (d) shows the distribution of points in each cluster. These results demonstrate that the methods we propose offer utility beyond naïve hierarchical or k-means clustering and are better suited to recover patterns in neural data.

Clarify the paper's goals and contributions:– As noted above, the use of an unsupervised pipeline to map spiking activity is not novel; the survey of data is interesting, but purely descriptive. The key contributions were then not obvious to this reader – please clarify. To give some suggestions, while reading 3 possible contributions came to mind, but none quite fit: if the paper is an announcement of the dataset, then the datasets need a detailed explanation; if the paper is the introduction of a pipeline, then the pipeline ought to be fully unsupervised, else it is of little use outside the hands of the present lab; if the paper is about insights into the pyloric circuit, then conclusions and insights ought to be drawn from the descriptive resultsImprove the Discussion:– Explicitly link the results to insights into the pyloric circuit. The abstract states "strong mechanistically interpretable links between complex changes in the collective behavior of a neural circuit and specific experimental manipulations" – but only the short section on "Diversity and stereotypy in trajectories from functional to crash states" touches on this.

We have rewritten the entire Discussion. A new section (Linking circuit outputs to circuit mechanisms) reviewers links between our results and what we learn about the pyloric circuit.

– Long section on other clustering methods is a survey of some alternatives and draws no conclusions about why tSNE was chosen here: either use to justify tSNE, or omit.

We have removed this section.

– Lines 678–716 are not based on any results in the paper

We have removed this section.

Discuss limitations:– The omission of the PY neuron activity means that the map as given is incomplete – what might it look like with it (new states, new transitions etc)?

We agree with the reviewer that the omission of the PY neurons’ activity means that the map is incomplete. There are likely many more states, and hence many more transitions, than the ones we have identified. In addition, we note that there are other pyloric neurons whose activity is also missing (AB, IC, LPG, VD). However, measuring just LP and PD allows us to monitor the activity of the most important functional antagonists in the system (because they are effectively in a half-center oscillator because PD is electrically coupled to AB). In general, the more neurons one measures, the richer the description of the circuit dynamics will be. Collecting datasets at this scale (~500 animals) from all pyloric neurons is challenging, and we have revised the manuscript to make this important point (see Discussion: Technical considerations).

– The transitions are slow and discrete by design (20 s long segments), whereas in reality they may be abrupt, or continuous.

We agree with the reviewer. There are tradeoffs in choosing a bin size in analyzing time series – choosing longer bins can increase the number of “states” and choosing shorter bins can increase the number of transitions. We chose 20s bins because it is long enough to include several cycles of the pyloric rhythm, even when decentralized, yet was short enough to resolve slow changes in spiking. We have added a new section called “Technical considerations” to make this important point.

– tSNE's limitations and their implications e.g. it cannot capture hierarchical structure, nor has a null model to demonstrate that the underlying data contains some clustering structure.

This is an important point. We have added this to a new section in the Discussion called “Technical considerations”.

– Why those features of the spike–trains, and how would the map change if features were omitted or added? (e.g. the regularity of spiking).

We have repeated the embedding deleting randomly chosen columns in our matrix of feature vectors. This gives us some insight into whether our results are strongly dependent on any one feature. In the following figure, each panel shows the result of embedding following the deletion of a randomly chosen column in the feature matrix. All coloring is from the original map. All maps look similar, and gross features of the map are preserved, suggesting that the map does not depend sensitively on any one feature.

Reviewer #3:Gorur–Shandilya et al. apply an unsupervised dimensionality reduction (t–SNE) to characterize neural spiking dynamics in the pyloric circuit in the stomatogastric ganglion of the crab. The application of unsupervised methods to characterize qualitatively distinct regimes of spiking neural circuits is very interesting and novel, and the manuscript provides a comprehensive demonstration of its utility by analyzing dynamical variability in function and dysfunction in an important rhythm–generating circuit. The system is highly tractable with small numbers of neurons, and the study here provides an important new characterization of the system that can be used to further understand the mapping between gene expression, circuit activity, and functional regimes. The explicit note about the importance of visualization and manual labeling was also nice, since this is often brushed under the rug in other studies.While the specific analysis pipeline clearly identifies qualitatively distinct regimes of spike patterns in the LP/PD neurons, it is not clear how much of this is due to t–SNE itself vs the initial pre–processing and feature definition (ISI and spike phase percentiles). Analyses that would help clarify this would be to check whether the same clusters emerge after (1) applying ordinary PCA to the feature vectors and plotting the projections of the data along the first two PCs, or (2) defining input features as the concatenated binned spike rates over time of the LP & PD neurons (which would also yield a fixed–length vector per 20 s trial), and then passing these inputs to PCA or t–SNE. As the significance of this work is largely motivated by using unsupervised vs ad hoc descriptors of circuit dynamics, it will be important to clarify how much of the results derive from the use of ISI and phase representation percentiles, etc. as input features, vs how much emerge from the dimensionality reduction.

We agree with the reviewer that is important to clarify how much of our results come from the data itself, and how we parameterize them using ISIs and phases, and how much comes from the choice of t-SNE as a dimensionality reduction algorithm. We have addressed this concern in the following ways:

1. We used principal components analysis on the feature vectors and measured triadic differences in features such as the period and duty cycle of the PD neuron. We found that triadic differences were lower in the t-SNE embedding than in the first two PCA features, or in shuffled t-SNE embeddings (Figure 2– Figure supplement 2), suggesting that the embedding is creating a useful representation that captures key features of the data.

2. We have used uMAP to reduce the dimensionality of the feature matrix to two dimensions and found that it too preserved the coarse features of the embedding that we observe with t-SNE. Coloring the uMAP embedding by the t-SNE labels revealed that the overall classification scheme was intact (Fig 3 – figure supplement 3).

3. We generated a synthetic dataset and applied the unsupervised part of our algorithm to it (conversion to ISIs, phases, etc., then t-SNE). We colored the points in the t-SNE embedding by the category in the synthetic dataset. We found that categories were well separated in the t-SNE plot, and each cluster tended to have a single color. This validates the overall power of our approach and shows that it can recover clustering information in large spike sets (Figure 2—figure supplement 4).

4. We have run k-means and hierarchical clustering on the feature vectors directly and shown that our method is superior to these naïve clustering algorithms running on the feature vectors. We speculate that this is because these clustering methods attempt to partition the full space using global distances, at the expense of distance along the manifold on which the data is located. Algorithms like t-SNE are biased towards local distances, and discount global distances between points outside a neighborhood, and are this better suited here.

Please define all acronyms when they are introduced.

Fixed.

Why is 20 seconds chosen as the time window to compute ISIs/phase representations before feeding the data into t–SNE? Is this roughly the window one needs to look at to visually identify differences across dynamical regimes and can the authors explain why?

Binning and analyzing data in bins is routine in any time-series analysis. 20s bins were chosen to be large enough that there are rich spike patterns within the bin, and small enough so that long-term changes could be visualized as changes in states. Smaller bins increase the number of points that need to be embedded. We have added a statement about this to the section titled Discussion: Technical considerations.

It would be useful to note in the manuscript that t–SNE is tuned to preserve local over global similarities, which gives it its utility in clustering. It would also be useful to discuss the relationship of t–SNE to UMAP (McInnes, Healy, Melville 2018 arXiv: https://arxiv.org/abs/1802.03426), another popular dimensionality reduction technique in neuroscience.

We have revised the manuscript to include using uMAP to perform the dimensionality reduction instead of t-SNE (Figure 3 – figure supplement 3). As we pointed out in the original manuscript (lines 885), recent work (Kobak and Linderman 2021) has found that t-SNE and uMAP are more similar than previously thought, with differences arising substantially from differences in initialization.

Most uses of t–SNE in biology in the past have been in the context of classifying behavior (e.g. Berman et al. 2014 J R Soc Interface: https://www.ncbi.nlm.nih.gov/pmc/articles/PMC4233753/ or Clemens et al. Current Biology 2018: https://www.sciencedirect.com/science/article/pii/S0960982218307735). Including these references would be helpful as a point of comparison and to emphasize the novelty of applying t–SNE to neural spiking data.

We cite Berman *et al.* and other relevant papers for these reasons (line 551 in original manuscript).

Can the authors comment on how much pyloric rhythms can deviate from the standard triphasic pattern before behavior (or digestion?) is significantly disrupted?

Unfortunately, there are very few studies of motor patterns in intact animals, and it would be very hard to know how much real disruption of food movement would occur, even if the pyloric rhythm were somewhat disrupted. So sadly, we have no idea how to answer this question. That said, the pyloric rhythm is remarkably reliable in vivo under normal conditions.

In Figure 6 it would be helpful to show example time–series going into the transition matrix analysis, with identified state labels at the different timepoints. It wasn't clear whether timepoints were e.g. moment–to–moment, 20 sec chunks, a 20 sec sliding window, etc.

Time series of states are shown in Fig 5—figure supplement 1. Transition analysis was performed taking care only to measure transitions in contiguous chunks of data. All analysis in the manuscript uses 20s non-overlapping bins. We have added a sentence clarifying this in the Results.

In Figure 8 it would be helpful to show examples of what bursts look like with and without decentralization, in order to contextualize the quantitative metrics.

We have included a new figure supplement (Figure 8 – figure supplement 1) showing example traces before and after decentralization.

It would also be useful to connect this work in the discussion to modern approaches for identifying model parameter regimes underlying distinct neural activity patterns (see e.g. Bittner et al. 2021 bioRxiv https://www.biorxiv.org/content/10.1101/837567v3)

This is an interesting topic for further discussion, and is beyond the scope of our paper. In this manuscript we do not try to infer parameters of a mechanistic model that can generate these data; instead we focus on trying to understand the structure of the data and what stereotyped patterns we observe in it.

[Editors’ note: what follows is the authors’ response to the second round of review.]

The manuscript has been improved but there are some remaining issues that need to be addressed, as outlined below:Two reviewer concerns remain, which are explained more fully in the reviews below.1. The analysis pipeline outlined in this paper is rather ad hoc. This isn't a general "tool" as it has only been demonstrated in one particular biological system.

We agree with the reviewers that this isn’t a general “tool” and we have demonstrated the utility of this approach only in one biological system. We have revised all descriptions of our approach to remove the word “tool”. Even though we have limited ourselves to a single circuit type, we have shown that our approach allows us to ask many interesting questions of the system and refer the reader to the Discussion section where we talk about its applicability to other systems.

2. It is acceptable for the authors to call the core method (tSNE) unsupervised. The "manual curation" is part of any exploratory data analysis, though it is generally preferable to for this manual curation / interpretation step to be as objective and reproducible as possible (which is arguably not entirely achieved in this paper). The paper can be seen as an investigation into how to apply an existing tool (tSNE) to a very well-studied and simple neural system.In revision, we ask the authors to take a final pass through the Introduction and Discussion and to tone down any comments about this being a general, off-the-shelf tool and to note that while the core algorithmic method (tSNE) is unsupervised, there is still the need for expert interpretation of the output.

We have carefully revised the Introduction and Discussion and removed any statements suggesting that our approach is an off-the-shelf tool and have revised its description to be explicit about what we did. We have explicitly stated that our approach requires manual annotation and classification of spike patterns (Introduction lines 68-71; Results – Visualization of circuit dynamics allows manual labelling and clustering of data; Discussion lines 494, 531, 623-631, 676-684).

Reviewer #1:The manuscript has been revised along the lines suggested in both the consensus suggestions and the individual reports. The rewritten Discussion is considerably improved. There is a lot of hard work here, and I've no doubt the results are useful to the PI's lab, and presumably other workers on this CPG.To my mind the authors did not really address two of the main concerns of the reviewers: (1) what the purpose of this paper is and (2) the validity of the manual clustering of tSNE. In more detail:Purpose of the paper: the paper prominently refers to it introducing an "unsupervised" method/approach (e.g. lines 69-70) and a "tool" (lines 537-541, 711-714). Neither are true: the approach here is by definition not unsupervised as it uses manual curation to identify the clusters upon which all further analyses are based; such manual curation means that the only "tool" parts are using tSNE on a feature vector, which doesn't merit the term.

We agree with the reviewer. We have removed all references to “unsupervised” in isolation and made it clear that t-SNE is an unsupervised method, but what we have done requires manual identification and labelling of clusters.

We further agree that the use of “tool” isn’t warranted. We have removed “tool” from all references to what we did and rephrased to make it clear that what we did uses both an off-the-shelf tool (t-SNE) and expert annotation.

Rather, it seems that the purpose of the paper is instead to provide a cohesive overview of the dynamics of this specific CPG system, by finding a way to systematically combine a large number of separate recordings from baseline and perturbed preparations, thus allowing knowledge discovery (e.g. the types of transitions; the changes in the regular state that precede transitions) on those combined data.

We agree with the reviewers’ assessment of our paper. This is indeed the purpose of this paper, and we hope it is clearer to the reader now that we have focused our attention on this.

tSNE and validating the clustering: the authors did considerable work related to this issue, by using their approach on synthetic data, replacing tSNE with UMAP, and by testing simple hierarchical and k-means clustering of their feature vectors. But none of these spoke directly to the validity of the clustering: the synthetic data and UMAP were still manually clustered, so could be made to conform to the desired results;

We apologize for not being clear – we did not manually re-cluster either the synthetic data or the UMAP embedding. We used the labelling generated in the original t-SNE map to color all other maps (t-SNE embeddings with different parameters, initializations, UMAP embeddings). The fact that these other maps appear “clustered” even when we use the original labelling offers (qualitative) evidence that the heavy lifting in the labelling isn’t being done by the human but by the dimensionality reduction technique we used. Similarly, we did not re-color or label clusters in the synthetic data embedding (Figure 2—figure supplement 5). The fact that clusters are almost entirely of a single color suggests that the first part of our method (feature reduction + t-SNE) is powerful enough that separating clusters in the synthetic data is largely trivial by a human.

The unsupervised clustering approach used shows the well-known problem that directly clustering a high-dimensional dataset (a 48-dimensional feature vector) is often impossible using classical techniques simply due to the curse of dimensionality. The normal approach here would be to use dimension reduction then unsupervised clustering (e.g. in the classic combination of PCA then k-means), matching the way the two steps are done in the present paper.

We thank the reviewer for this suggestion. We have added a new figure (Figure 2—figure supplement 2) showing the first two principal components of the feature vectors. We applied k-means to the first ten principal vectors and colored the dots in both the projection of the first two principal components and the t-SNE embedding using this k-means clustering, demonstrating the point the reviewer makes.

As the paper shows, the clusters found in tSNE by manual curation are arbitrary. For example, changing the perplexity parameter changes the fragmentation of the tSNE map, so would likely lead to different manual clusterings.

Figure 2—figure supplement 4 shows the opposite. We use the same coloring for all perplexities, and we see that as long as the perplexity is sufficiently high, we get qualitatively the same result. If one were to manually re-cluster embeddings at higher perplexities, the result would be similar.

In another example, the clusters found by manual curation are not all contiguous in the 2D space e.g. the LP-silent cluster is in 3 groups, one of which is on the opposite side of the 2D space to the others – so distance in this space is not interpreted as meaningful for all clusters.

This is correct. We previously reasoned that most of the distance measurements were between pairs of points within the same cluster, but we agree with the reviewer that we did not rigorously show this. We have removed all distance-based metrics and plots from the paper, because it is confounded by this effect.

In another example, the dividing line between some "clusters" is arbitrarily put in a contiguous run of points – e.g. the LP-weak-skipped (pink) and irregular-bursting (brown).

This is correct. This stems from a smooth variation in observed spike patterns between one canonical form to another, so there truly is no discontinuity when the raw spike patterns are observed. This can be seen clearly in our interactive visualization of the data (a preview of which is available at https://srinivas.gs/stg). When putting smoothly varying data into categorical bins, this is unavoidable. A strength of our visualization method is that it is clear when there exist clear distinctions between categories (e.g., between PD-weak-skipped and LP-weak-skipped) and when there does not (e.g., between the LP-weak-skipped and irregular bursting, as the reviewer points out).

Reviewer #3:One remaining edit that would be helpful is to show the projection of the data onto the first two PCs (re: the PCA analysis referenced in Figure 2-Figure supp 2), so that the reader can visualize what structure there is prior to computing the triadic difference statistics.

We show the first two principal components in the new Figure 2 —figure supplement 2.

If not too computationally intensive, it would also be useful to show what emerges from applying a naive clustering or visualization algorithm directly on the spike-train time-series [see (2) in Reviewer #3 major concern], even if only to show that it is quite messy.

There is no straightforward way to do this. If we treat the spike trains as binary series, small shifts in the time of first spike can lead to spurious large differences between similar or identical spike trains. This was our motivation for converting spike trains to ISI sets. Even then, because ISI sets are of arbitrary length, it is not trivial to directly apply a naïve clustering or visualization algorithm.